# Exploring FDA-Approved Frontiers: Insights into Natural and Engineered Peptide Analogues in the GLP-1, GIP, GHRH, CCK, ACTH, and α-MSH Realms

**DOI:** 10.3390/biom14030264

**Published:** 2024-02-22

**Authors:** Othman Al Musaimi

**Affiliations:** 1School of Pharmacy, Faculty of Medical Sciences, Newcastle University, Newcastle upon Tyne NE1 7RU, UK; othman.almusaimi@newcastle.ac.uk; 2Department of Chemical Engineering, Imperial College London, London SW7 2AZ, UK

**Keywords:** peptides, FDA, natural peptides, GLP-1, GIP, GHRH, GH, CCK, ACTH, α-MSH, diabetes, insulin, oxytocin, blood pressure, vasoconstrictor, Cushing’s disease, hypoglycemia, premenopausal

## Abstract

Peptides continue to gain significance in the pharmaceutical arena. Since the unveiling of insulin in 1921, the Food and Drug Administration (FDA) has authorised around 100 peptides for various applications. Peptides, although initially derived from endogenous sources, have evolved beyond their natural origins, exhibiting favourable therapeutic effectiveness. Medicinal chemistry has played a pivotal role in synthesising valuable natural peptide analogues, providing synthetic alternatives with therapeutic potential. Furthermore, key chemical modifications have enhanced the stability of peptides and strengthened their interactions with therapeutic targets. For instance, selective modifications have extended their half-life and lessened the frequency of their administration while maintaining the desired therapeutic action. In this review, I analyse the FDA approval of natural peptides, as well as engineered peptides for diabetes treatment, growth-hormone-releasing hormone (GHRH), cholecystokinin (CCK), adrenocorticotropic hormone (ACTH), and α-melanocyte stimulating hormone (α-MSH) peptide analogues. Attention will be paid to the structure, mode of action, developmental journey, FDA authorisation, and the adverse effects of these peptides.

## 1. Introduction

Peptides represent medium-sized molecules, situated between small molecules and large antibodies [1]. Furthermore, they have proven to be superior alternatives to both classes. For example, peptides surpass small molecules in terms of high specificity and the capacity to inhibit extensive surface interactions. Additionally, they display lower immunogenicity, and their production is more cost-effective compared to that of large biologics [2,3]. Approximately 100 peptide drugs are currently in use in the global market, and there are hundreds of new peptides undergoing clinical development and preclinical studies [3]. Classical solution peptide synthesis (CSPS) is the first peptide synthesis methodology used to prepare short and long peptides with good purity and yields [4,5]. Despite its environmentally friendly approach, which involves using minimal reagent amounts, it is perceived as a laborious process due to the necessity for separating and characterising intermediates. Solid-phase peptide synthesis (SPPS) is widely regarded as the preferred method for producing peptides at both research and industrial levels. Liquid-phase peptide synthesis (LPPS), on the other hand, combines the strengths of both classical solution peptide synthesis CSPS and SPPS. In LPPS, a soluble tag, distinct from the reagents and byproducts, is utilised for peptide growth. This method enables the utilisation of smaller amounts of reagents compared to what is required in SPPS [6].

Peptides face challenges related to low stability and susceptibility to degradation by enzymatic peptidases, leading to a relatively short half-life in vivo [7,8]. Consequently, multiple strategies have been explored to improve the stability of peptides and facilitate their entry into the market [1,9]. Cyclisation is a prevalent chemical modification employed to restrict the structure of peptides, enhancing their interaction with targets and cellular permeability. This approach encompasses diverse strategies, including head-to-tail, backbone-to-sidechain, and sidechain-to-sidechain cyclisation [10,11,12]. Substituting nonproteinogenic amino acids has proven to be a successful chemical modification strategy for enhancing the proteolytic stability of peptides [13,14]. *N*-methylation [13,15] and substitution with β-amino acids [16] serves the same purpose of enhancing the proteolytic stability of peptides.

Since 1923, the FDA has approved approximately ten natural peptides for various indications, nine peptides for diabetes, four growth hormone-releasing hormones (GHRH), one cholecystokinin (CCK), two adrenocorticotropic hormones (ACTH), and three analogues of α-melanocyte-stimulating hormone (α-MSH) peptides (FDA) (Figure 1).

In this review, I will shed light on the chemical structures and properties, the developmental journey, the therapeutic target, the mode of action, and FDA approval, concluding with remarks regarding these peptides.

## 2. Natural Peptides

The field of drug discovery has increasingly embraced the use of synthetic peptides derived from natural sources. Synthetic analogues have demonstrated the ability to exert similar therapeutic effects to those of their endogenous counterparts. This represents a crucial advancement for meeting the growing demands for these entities. The FDA has approved ten therapeutic peptides that draw inspiration from their endogenous analogues, further emphasising the recognition and acceptance of synthetic peptides in therapeutic applications (Table 1).

### 2.1. Insulin (Iletin)

Insulin stands as the inaugural isolated natural peptide, serving as a peptide hormone produced and released by the β-cells located in the pancreatic islets [17]. Insulin was isolated by Frederick Banting and Best in 1921 [17,18,19]. Insulin is a 51-mer acid peptide consisting of two chains, A and B, linked by three disulfide bridges. These bridges connect Cys6 and Cys11 of chain A, Cys7 of chain A and Cys7 of chain B, and Cys20 of chain A and Cys19 of chain B (Figure 2). In 1922, insulin became the first therapeutic peptide approved for treating patients with diabetes mellitus. Eli Lilly and Company commercialised it in 1923 [19,20]. It is used to enhance glycemic control in both adults and children diagnosed with diabetes mellitus [21].

Insulin binds to the α subunit of the insulin receptor (INSR) located on muscle and fat cells. Upon insulin binding to the extracellular domain of the α subunits, the receptor undergoes conformational changes that activate the intrinsic tyrosine kinase activity of the β subunits. This activation leads to autophosphorylation of the β subunits and subsequent phosphorylation of downstream signalling molecules, initiating a cascade of events that regulate various cellular processes such as glucose uptake and metabolism (Figure 3) [22]. Signalling events downstream of INSR activation can be broadly categorised into mitogenic (related to cell growth and division) and metabolic signals [23,24,25]. INSR mediates a variety of cellular responses, coordinating both growth-related and metabolic processes [22]. As a result, insulin works to lower blood glucose levels by facilitating the cellular uptake of glucose and simultaneously inhibiting the release of glucose from the liver [21,22].

Eli Lilly and Company developed insulin, and it received FDA approval in 1923 [21]. Eli Lilly and Company also developed insulin lispro (Humalog), which distinguishes itself from human insulin by the substitution of lysine for proline at position B28 and proline for lysine at position B29 in its amino acid sequence [26]. It received FDA approval in 1996 [26]. Additionally, they developed insulin aspart (NovoLog), structurally similar to regular human insulin, differing only by a single substitution in which the amino acid proline is replaced by aspartic acid at position B28. This replacement in NovoLog diminishes the molecule’s inclination to form hexamers, a characteristic observed with regular human insulin. Consequently, NovoLog is absorbed more rapidly after subcutaneous injection when compared to regular human insulin [27]. It received FDA approval in 2000 [27]. It is typically administered subcutaneously and may be associated with adverse effects such as allergic reactions, local injection site reactions, lipodystrophy, rash, and pruritus [21].

### 2.2. Corticotropin (H.P. Acthar Gel)

In 1954, corticotropin was isolated from the anterior pituitary of hogs by Bell et al. at Stamford Laboratories, and its tentative structure was deduced [28]. In this study, the sequence of the initial 24 amino acids from the amino end was determined, along with the first 11 amino acids from the *C*-terminal end [28,29]. In 1955, the same research group deduced the complete sequence of corticotropin (Figure 4) [29].

Corticotropin is a 41-mer acid peptide (H-SQEPPISLDLTFHLLREVLEMTKADQLAQQAHSNRKLLDIA-OH) that plays a role in mediating both endocrine and autonomic responses to stress (Figure 4) [30,31]. Corticotropin is utilised in the treatment of various specific and inadequately defined steroid-responsive disorders. Additionally, it is used in the management of multiple sclerosis and the treatment of infantile spasms in infants and children under 2 years of age [32,33].

Two receptors for corticotropin-releasing hormone (CRH) exist, namely CRH-R1 and CRH-R2. When CRH binds to either CRH-R1 or CRH-R2, the receptors undergo structural changes facilitating their interaction with and activation of Gs proteins. The activated Gs proteins then stimulate adenylate cyclase, initiating the conversion of adenosine triphosphate (ATP) to cyclic adenosine monophosphate (cAMP). Elevated cAMP levels function as a secondary messenger, initiating diverse intracellular signalling pathways. One prominent downstream effector of cAMP is protein kinase A (PKA). Ultimately, these signalling pathways orchestrate various physiological responses, including the regulation of stress response, metabolism, and immune function. [34]. Optimal effects of a trophic hormone on a target organ are attained when the hormone is continuously present in optimal amounts. Consequently, a fixed dose of H.P. Acthar Gel is expected to exhibit a linear increase in adrenocortical secretion as the duration of the infusion increases [32].

Corticotropin received FDA approval in 1952, and later, in 2001, Questcor Pharmaceuticals acquired the rights to the drug [33]. It is administered subcutaneously and intramuscularly and may lead to various adverse reactions. These include fluid retention, alterations in glucose tolerance, elevation in blood pressure, behavioural and mood changes, increased appetite, and weight gain. Additionally, specific effects observed in children under 2 years old include an increased risk of infections, hypertension, irritability, Cushingoid symptoms, cardiac hypertrophy, and weight gain [32]. The plasma half-life of H.P. Acthar Gel is approximately 15 min. However, the pharmacokinetics of H.P. Acthar Gel have not been thoroughly characterised [32].

### 2.3. Cyclosporine (Sandimmune)

Cyclosporine is an 11-mer cyclic peptide and a calcineurin inhibitor (antagonist) that effectively inhibits T cell activation [35]. The cyclic structure, along with the incorporation of *N*-methyl and unusual amino acids within its sequence, imparts improved stability and pharmacokinetics to the peptide (Figure 5) [36]. Cyclosporine is utilised as an immunosuppressant agent [37].

Cyclosporine binds to the receptor cyclophilin-1 inside the cells, forming a complex known as cyclosporine–cyclophilin. This complex inhibits calcineurin, preventing the dephosphorylation and activation of the nuclear factor of activated T cells (NF-AT), which plays a crucial role in inflammatory reactions (Figure 6) [35,38,39]. Cyclosporine exhibits a preference for inhibiting T-lymphocytes, with the T-helper cell being the primary target, although T-suppressor cells may also be affected. Additionally, it inhibits the production and release of lymphokines, including interleukin-2 [37].

Cyclosporine is primarily distributed outside the blood volume. The steady-state volume of distribution, reported as 3–5 L/kg in solid organ transplant recipients during intravenous dosing, reflects its distribution in various body compartments. In blood, distribution is concentration-dependent, with approximately 33–47% in plasma, 4–9% in lymphocytes, 5–12% in granulocytes, and 41–58% in erythrocytes. At high concentrations, leukocytes and erythrocytes reach binding capacity saturation. In plasma, around 90% of cyclosporine binds to proteins, mainly lipoproteins. Notably, cyclosporine is excreted in human milk [37].

The discovery and initial development of cyclosporin occurred in the 1970s through the work of Borel et al. [40,41]. Cyclosporine was developed by Sandoz (now Novartis) and received FDA approval in 1983 [42]. It was initially marketed under the brand name Sandimmune. It is administered orally and has been associated with various adverse effects, including renal dysfunction, tremor, hirsutism, hypertension, gum hyperplasia, headache, gastrointestinal disturbances, and hirsutism/hypertrichosis [37]. The disposition of cyclosporine from the blood follows a generally biphasic pattern, with a terminal half-life of approximately 8.4 h (with a range of 5–18 h) [37]. Cyclosporine undergoes extensive metabolism, primarily by the cytochrome P-450 3A enzyme system in the liver, with lesser contributions from the gastrointestinal tract and the kidneys [37].

### 2.4. Oxytocin (Syntocinon)

In 1953, utilising the partial hydrolysis approach, the Nobel Laureate du Vigneaud was the first to propose the structure of oxytocin (Figure 7) [43]. One year later, Vigneaud also reported the first synthesis of oxytocin and a cyclic octapeptide amide with hormonal activity [44,45]. The same group also proposed alternative synthetic approaches for oxytocin [46].

Oxytocin is a 9-mer peptide amide (H-CYIQNCPLG-NH2) that contains a disulfide bridge within its structure (Figure 7). It serves as the primary hormone for inducing uterine contractions and facilitating milk ejection from the posterior pituitary gland [45]. The biological activity of oxytocin was confirmed through assays that included the avian depressor effect, inducing contractions in the rat uterus (uterine-contracting activity), and facilitating the ejection of milk [47].

Studies have indicated that the terminal amino group plays a crucial role in the activity stimulation process of oxytocin. Additionally, the hydroxyl group of Tyr is essential for receptor binding and contributes, to a lesser extent, to the overall activity of oxytocin [48]. Various analogues were investigated, revealing that blocking the terminal amino group resulted in the inhibition of oxytocin activity. However, these analogues still demonstrated binding capability [48].

Oxytocin binds to a membrane receptor coupled to a G protein. The resulting complex activates phospholipase Cβ (PLCβ), leading to the hydrolysis of phosphatidylinositol biphosphate (PIP2) and the generation of inositol triphosphate (IP3) and diacylglycerol (DAG). IP3 triggers the release of calcium from the endoplasmic reticulum and facilitates the influx of extracellular calcium by inducing conformational changes in voltage-operated calcium channels. Subsequently, the intracellular calcium binds to calmodulin, activating the myosin light chain kinase enzyme (MLCK). MLCK phosphorylates the myosin light chain (MLC), initiating the mechanism for myometrial contraction. Additionally, oxytocin can elevate intracellular calcium levels through the mitogen-activated protein kinase (MAPK) pathway. This activation induces the expression of the cyclo-oxygenase II isoform (COX-II), leading to the transformation of arachidonic acid into prostaglandins (Figure 8) [49,50].

Parke Davis Pharmaceutical received FDA approval for their synthetic oxytocin in 1996 [52]. It is marketed under various names, including Pitocin, Syntocinon, and Viatocinon. Oxytocin is administered intravenously and may have adverse effects, including anaphylactic reaction, postpartum haemorrhage, cardiac arrhythmia, fatal afibrinogenemia, nausea, vomiting, premature ventricular contractions, and pelvic hematoma [53].

### 2.5. Glucagon (Baqsimi)

Glucagon is a 29-mer linear acid peptide that shares the same sequence with the human glucagon (H-HSQGTFTSDYSKYLDSRRAQDFVQWLMNT-OH) (Figure 9). It is used to manage and treat hypoglycaemia, serving as an antidote to beta-blocker and calcium channel blocker overdose, addressing anaphylaxis refractory to epinephrine, and aiding in the passage of food boluses [54]. 

Glucagon binds to the glucagon receptor, activating Gsα and Gq. This activation leads to the stimulation of adenylate cyclase, resulting in an increase in intracellular cAMP. Subsequently, cAMP activates PKA, which phosphorylates glycogen phosphorylase. Phosphorylated glycogen phosphorylase then phosphorylates glycogen, initiating its breakdown and releasing glucose into the bloodstream (Figure 10) [55]. Following subcutaneous administration, the maximum plasma concentration of glucagon (7.9 ng/mL) is reached within 20 min. In contrast, intramuscular administration yields a maximum concentration of 6.9 ng/mL within 13 min [54].

Glucagon was developed by Eli Lilly and Company and received FDA approval in 1998 [54]. Glucagon is administered intravenously, intramuscularly, or subcutaneously. It may be associated with side effects such as difficulty breathing, loss of consciousness, and the development of a rash with scaly, itchy, red skin on the face, groin, pelvis, or legs [54]. The half-life of glucagon is relatively short, ranging from 8 to 18 min [54].

### 2.6. Secretin (ChiRhoStim)

Secretin is a gastrointestinal hormone produced by cells in the duodenum in response to acidification [56]. It is a 27-mer peptide amide (H-HSDGTFTSELSRLREGARLQRLLQGLV-NH_2_) (Figure 11). Secretin is used for the regulation of gastric acid, the regulation of pancreatic bicarbonate, and osmoregulation [56].

Secretin binds to the basolateral membranes of ductular and duct cells in the pancreas and increases the levels of cAMP, leading to the phosphorylation of PKA and activation of the cystic fibrosis transmembrane conductance regulator (CFTR). The activation of CFTR, in turn, stimulates the Cl^−^/HCO_3_^−^ anion exchanger 2, resulting in the secretion of bicarbonate-rich pancreatic fluid. Additionally, secretin promotes the secretion of water and electrolytes in cholangiocytes [57,58]. It mediates an inhibitory effect on acid secretion by parietal cells of the stomach. Simultaneously, it stimulates the release of pancreatic juice, which contains high amounts of water and bicarbonate ions. This leads to the alkalinisation of the duodenal content (Figure 12) [56]. Following intravenous administration of 0.4 mcg/Kg, synthetic human secretin concentration swiftly returns to baseline secretin levels within 90 to 120 min [56].

Synthetic secretin was developed by ChiRhoClin and received FDA approval in 2002 [60]. It is administered by intravenous injection, and it may be associated with adverse effects such as allergic reactions, including skin rash; itching or hives; swelling of the face, lips, or tongue; breathing problems; dizziness; feeling faint or lightheaded; falls; and a slow or irregular heartbeat [56]. The elimination half-life is approximately 45 min, with a clearance of 580.9 ± 51.3 mL/min and a volume of distribution of 2.7 L [56].

### 2.7. Calcitonin (Miacalcin)

Calcitonin is a 32-mer peptide amide (H-CSNLSTCVLGKLSQELHKLQTYPRTNTGSGTP-NH_2_) that contains a disulfide bridge between Cys1 and Cys7 (Figure 13).

Calcitonin plays a role in controlling the level of calcium in the blood. It is produced by the parafollicular cells (C cells) in the thyroid gland. The analgesic activity of calcitonin, which can exhibit both rapid onset and sustained effects, may be related to its secretion from the thyroid gland [61]. The analgesic activity of calcitonin involves two synergistic mechanisms: central and peripheral [61]. The analgesic activity of calcitonin involves interaction with the serotoninergic and catecholaminergic systems, affecting specific central nervous system (CNS) receptors (central). Additionally, it includes the increase in the release of β-endorphin and the inhibition of prostaglandins (peripheral), as well as the modulation of calcium flux (central or peripheral) [61,62]. The binding sites for calcitonin are situated in the medial pontine reticular formation and the rostrocaudal axis of the periaqueductal grey matter. These CNS structures are integral parts of the pathways involved in the perception, transmission, and modulation of sensory stimuli [62,63]. For exogenous calcitonin to exert its analgesic effect, it must cross the blood–brain barrier, accumulate in these regions, and engage with receptors in the CNS [64]. The absolute bioavailability of calcitonin salmon is around 66% and 71% following intramuscular or subcutaneous injection, respectively. Subcutaneous administration leads to peak plasma levels within approximately 23 min [65].

Synthetic calcitonin was developed by Novartis and received FDA approval in 2005 [66]. It is administered either intravenously or intramuscularly. Some of the reported adverse effects include nausea, with or without vomiting (10%), injection site inflammation (10%), and flushing of the face or hands (2–5%) [65]. The terminal half-life is roughly 58 min for intramuscular administration and 59 to 64 min for subcutaneous administration [65].

### 2.8. Vasopressin (Vasostrict)

Vasopressin is a 9-mer peptide amide (H-CYFQNCPRG-NH_2_) with the same sequence as oxytocin, except Ile and Leu are replaced by Phe and Arg, respectively. It was first synthesized by the Du Vigneaud group (Figure 14) [67]. The biological activity of vasopressin was confirmed by data demonstrating an identical ratio of pressor to avian depressor activity as that of the natural hormone [67]. Vasopressin is prescribed to increase blood pressure in adults with vasodilatory shock who remain hypotensive after the administration of fluids and catecholamine therapy [68].

Vasopressin binds to its receptor through electrostatic interactions, where the positively charged vasopressin interacts with the negatively charged receptor. Short-range hydrogen bonding also plays a role in aligning vasopressin with its receptor site. Subsequently, a thiol–disulfide exchange reaction occurs, with the receptor mercaptide ion attacking the disulfide bridge of vasopressin. This leads to tertiary structure alterations and conformational changes in the protein components of the diffusion barrier. The resulting molecular sieve-like and loosened membrane allows the influx of water and other solutes. The vasopressin-receptor disulfide bond formed is eventually cleaved by reductase in a reverse reaction, restoring the membrane to its initial configuration (Figure 12, similar to the mechanism of secretin) [69]. Steady-state plasma concentrations are attained after 30 min of continuous intravenous infusion [68].

Par Pharmaceutical’s synthetic vasopressin, sold under the brand name Vasostrict, received FDA approval in 2014 [70]. In September 2021, the FDA approved Eagle’s abbreviated new drug application (ANDA) for vasopressin, and another version was launched in February 2022 by American Regent [70]. It is administered intravenously (IV) and may have adverse effects, including a decrease in cardiac output, bradycardia, tachyarrhythmias, hyponatremia, and ischemia (coronary, mesenteric, skin, digital) [68]. At infusion rates typically employed in vasodilatory shock (ranging from 0.01 to 0.1 units/minute), the clearance of vasopressin falls within the range of 9 to 25 mL/min/kg in patients with vasodilatory shock. The apparent half-life of vasopressin at these infusion levels is less than or equal to 10 min [68]. Vasopressin undergoes cleavage by serine protease, carboxypeptidase, and disulfide oxidoreductase at sites relevant to the hormone’s pharmacological activity. Consequently, the resulting metabolites are not anticipated to retain significant pharmacological activity [68].

### 2.9. Parathyroid Hormone (PTH) (Natpara)

Parathyroid hormone (PTH) is an 84-mer linear peptide amide (Figure 15) [71]. It is used as an adjunct to calcium and vitamin D to control hypocalcemia in patients with hypoparathyroidism [72].

PTH binds to the PTH receptor, increasing serum calcium levels through several mechanisms. These include enhancing renal tubular calcium reabsorption, increasing intestinal calcium absorption (by converting 25-hydroxy vitamin D to 1,25-dihydroxy vitamin D), and stimulating bone turnover, leading to the release of calcium into circulation (Figure 16) [71,72]. After single subcutaneous injections of PTH in individuals with hypoparathyroidism, peak plasma concentrations (mean t_max_) typically occur within 5 to 30 min, with a second, usually smaller, peak observed at 1 to 2 h [72].

PTH was developed by NPS Pharmaceuticals and received FDA approval in 2015 [73]. It is administered subcutaneously and may have adverse effects, including paraesthesia, hypocalcemia, headache, hypercalcemia, nausea, hypoaesthesia, diarrhoea, vomiting, arthralgia, hypercalciuria, and pain in the extremities [72]. The apparent terminal half-life was measured at 3.02 h for the 50 mcg dose and 2.83 h for the 100 mcg dose [72]. In vitro and in vivo investigations revealed that the hepatic process predominantly contributes to the clearance of parathyroid hormone, with a minor involvement of the kidneys [72].

### 2.10. Angiotensin II (Giapreza)

Angiotensin II is an 8-mer acidic peptide (H-DRVYIHPF-OH) and serves as the synthetic form of the endogenous angiotensin II (Figure 17). It is used as a vasoconstrictor to increase blood pressure in adults with septic or other distributive shock [74].

Angiotensin II binds to the G protein-coupled angiotensin II receptor type I on vascular smooth muscle. This interaction stimulates the Ca–calmodulin complex, resulting in the phosphorylation of MLC and initiating muscle contraction (Figure 8, same mechanism as for oxytocin) [74]. After intravenous infusion of angiotensin II in adults with septic or other distributive shock, the serum levels of angiotensin II remain comparable at baseline and 3 h after the infusion. However, after 3 h of treatment, the serum level of angiotensin I (the precursor peptide of angiotensin II) experiences a reduction of approximately 40% [74].

The development story of angiotensin II began in the latter part of the 19th century, with contributions from various research groups [75,76,77,78]. It took 70 years for angiotensin II to progress from its initial discovery to its marketing as a therapeutic molecule [79]. It was developed by La Jolla Pharmaceuticals and received FDA approval in 2017 [80]. It is administered intravenously, and thromboembolic events are reported as adverse effects [74]. The plasma half-life of intravenously administered angiotensin II is less than one minute [74].

## 3. Diabetes Peptide-Based Drugs

The FDA has authorized a variety of medications for the treatment of diabetes, each designed to target different receptors and operate through diverse mechanisms, depending on the specific type of diabetes being addressed (Table 2).

### 3.1. Diabetes Type I

#### 3.1.1. Insulin

Refer to natural peptides Section 2.1.

#### 3.1.2. Pramlintide (Symlin)

Pramlintide is a 37-mer peptide amide (KCNTATCATQRLANFLVHSSNNFGPILPPTNVGSNTY-NH_2_) that features a disulfide bridge between Cys2 and Cys7. It differs from the human amylin sequence by replacing Ala25, Ser28, and Ser29 with Pro in the corresponding three positions (Figure 18). It is used to treat type I and II diabetic patients treated with insulin [81].

Pramlintide functions as an amylinomimetic agent, mimicking the actions of the hormone amylin. This includes the modulation of gastric emptying, the inhibition of postprandial glucagon secretion, and the induction of satiety, ultimately resulting in decreased caloric intake and the potential for weight loss [81,82]. Pramlintide attaches to the calcitonin receptor’s core, along with one of the three receptor activity-modifying proteins, namely RAMP1, RAMP2, or RAMP3 [81,82]. The absolute bioavailability of a single subcutaneous dose is estimated to be approximately 30 to 40%. The injection administered in the arm resulted in higher exposure compared to injections in the abdominal area or thigh. [81].

Amylin Pharmaceuticals developed pramlintide, and it was granted FDA approval in 2005 [83]. Administered subcutaneously, it exhibited the following adverse effects: headache, dizziness, drowsiness, vision problems, hunger, weakness, sweating, confusion, irritability, a fast heart rate, and a jittery feeling [81,82]. In healthy subjects, the half-life of pramlintide is approximately 48 min, and it undergoes metabolism primarily in the kidneys [81].

### 3.2. Diabetes Type 2


**Glucagon-like peptide-1 (GLP-1) and glucose-dependent insulinotropic polypeptide (GIP).**


Both glucagon-like peptide-1 (GLP-1) and glucose-dependent insulinotropic polypeptide (GIP) are key incretins released from enteroendocrine cells in the gut [84]. GIP, a 42-mer peptide, originates from K cells in the upper intestine, while GLP-1, a 31-mer peptide, is produced by L cells in the lower intestine (Figure 19) [85,86]. 

GLP-1 and GIP regulate blood glucose levels associated with meals by maintaining a balance between insulin and glucagon [85,86]. Additionally, they inhibit gastric emptying, optimising nutrient absorption, and simultaneously managing weight gain [86]. GIP and GLP-1 bind to their respective receptors, initiating a cascade of events, including the activation of adenylate cyclase, the elevation of intracellular cAMP levels, and the activation of PKA. Furthermore, cAMP2 (EPAC2)/cAMP-guanine nucleotide exchange factor (GEF)II activates the exchange of protein. PKA plays a role in closing KATP channels, leading to membrane depolarization, and inhibits the delayed rectifying K^+^ (Kv) channel. This depolarization results in the opening of voltage-gated Ca^2+^ channels (VDCC), causing an increase in intracellular Ca^2+^ concentrations. The elevated Ca^2+^ concentrations eventually trigger the fusion of insulin-containing granules with the plasma membrane, facilitating insulin secretion from the β-cells. Additionally, this process promotes the transcription of the proinsulin gene, thereby increasing the insulin content within the β-cell. Notably, the activation of EPAC2 has been demonstrated to enhance the density of insulin-containing granules near the plasma membrane, further potentiating insulin secretion from the β-cell. ATP is involved in these intricate cellular processes (Figure 20) [85].

GLP-1 and GIP analogues share a structural resemblance of approximately 90% to the native GLP-1 and GIP, making them agonists capable of initiating the insulin production process (Table 2). Notably, certain GLP-1 analogues have demonstrated relevance in cardiovascular diseases. Patients administered with GLP-1 analogues exhibited lower incidences of cardiovascular-related deaths, nonfatal myocardial infarction, or nonfatal stroke [87].

#### 3.2.1. Exenatide (Byetta)

Exenatide is a 39-mer linear acid peptide (H-HGEGXFXSDLSKQMEEEAVRLFXEWLKNGGPSSGAPPPS-OH) that belongs to the GLP-1 family (Figure 21). It is used as an adjunct treatment to dietary and exercise measures to enhance glycemic control in adults diagnosed with type 2 diabetes mellitus [88].

Exenatide, a GLP-1 receptor agonist, binds to and activates the GLP-1 receptor, resulting in an augmentation of glucose-dependent insulin synthesis and the in vivo secretion of insulin from pancreatic β cells [88]. Similar to GLP-1 and other analogues, the mechanism involves raising the levels of cAMP, along with the activation of other downstream signalling pathways [88]. After subcutaneous administration to patients with type 2 diabetes, exenatide reaches median peak plasma concentrations in 2.1 h. Similar exposure levels are achieved with subcutaneous administration of exenatide in the abdomen, thigh, or upper arm [88].

Amylin Pharmaceuticals developed exenatide, and it received FDA approval in 2005 [89]. Administered subcutaneously, it has been associated with the following adverse effects: nausea, hypoglycaemia, vomiting, diarrhoea, jittery feelings, dizziness, headache, and dyspepsia. It is worth noting that nausea tends to decrease over time [88]. Exenatide is primarily eliminated through glomerular filtration, followed by proteolytic degradation. The mean apparent clearance of exenatide in humans is 9.1 L/h, and with a half-life of 2.4 h; it is recommended to be administered twice daily [88].

#### 3.2.2. Liraglutide (Victoza)

Liraglutide is a 31-mer peptide that belongs to GLP-1 family. It possesses a hexadecanoyl-Glu-containing moiety attached to its Lys side chain (Figure 22). It is used as an adjunct treatment alongside diet and exercise to enhance glycemic control in adults diagnosed with type 2 diabetes mellitus. Additionally, it is utilised to lower the risk of major adverse cardiovascular events in adults with type 2 diabetes mellitus and established cardiovascular disease [90].

Liraglutide, functioning as a GLP-1 receptor agonist, binds to and activates the GLP-1 receptor. This activation leads to an elevation in intracellular cAMP, thereby promoting insulin secretion when confronted with elevated glucose levels. Additionally, liraglutide exhibits the capability to reduce glucagon levels in a glucose-dependent manner. The overall process also encompasses a delay in gastric emptying [90]. The self-association and delayed absorption, coupled with its binding to plasma proteins and stability against metabolic degradation by dipeptidyl peptidase IV (DPP-IV) and neutral endopeptidase (NEP), contribute to a pharmacokinetic profile that renders it well-suited for once-daily administration [90]. Following subcutaneous administration, maximum concentrations of liraglutide are achieved at 8–12 h post dosing [90].

Liraglutide was developed by Novo Nordisk and received FDA approval in 2010 [91]. Administered subcutaneously, it may lead to the following adverse effects: nausea, diarrhoea, vomiting, decreased appetite, dyspepsia, and constipation [90]. In the first 24 h after the administration of a single dose of liraglutide to healthy subjects, the primary component in plasma was intact liraglutide. Liraglutide undergoes endogenous metabolism in a manner similar to that of large proteins, with no specific organ identified as a major route of elimination [90].

#### 3.2.3. Lixisenatide (Adlyxin)

Lixisenatide is a 44-mer peptide amide (H-HGEGTFTSDLSKQMEEEAVRLFIEWLKNGGPSSGAPPSKKKKKK-NH_2_) that belongs to the GLP-1 family (Figure 23). Lixisenatide was derived from exenatide, with a modification involving the replacement of a Pro residue from the *C*-terminal with a linker comprising six Lys residues [92]. With its six Lys residues, it exhibits a heightened resistance to degradation by DPP-IV, thereby extending its half-life and facilitating once-daily dosing [93].

It is used as an adjunct treatment alongside diet and exercise to enhance glycemic control in adults diagnosed with type 2 diabetes mellitus [94].

Lixisenatide, functioning as a GLP-1 receptor agonist, binds to and activates this receptor, thereby stimulating adenylyl cyclase. This activation results in increased glucose-dependent insulin secretion, decreased glucagon secretion, and a slowing of gastric emptying [86,94].

It was developed by Sanofi Aventis and approved by the FDA in 2013 [95]. Administered subcutaneously, it may elicit the following side effects: nausea, vomiting, headache, diarrhoea, dizziness, and hypoglycaemia [94]. Lixisenatide was discontinued in 2023, with the decision attributed solely to business considerations and not related to any concerns about the safety or efficacy of the drug [96].

#### 3.2.4. Albiglutide (Tanzeum)

Albiglutide consists of two copies of a 30-mer linear acid peptide (H-HGEGTFTSDVSSYLEGQAAKEFIAWLVKGRG-OH), and it belongs to GLP-1 family (Figure 24). It is used as an adjunct treatment alongside diet and exercise to enhance glycemic control in adults diagnosed with type 2 diabetes mellitus [97].

Abiglutide, functioning as a GLP-1 receptor agonist, activates the GLP-1 receptor, leading to increased glucose-dependent insulin secretion, and it also has the effect of slowing gastric emptying [98]. 

It was developed by GSK and received FDA approval in 2014 [99]. However, in 2017, it was discontinued primarily due to limited prescribing of the drug and not because of any safety or efficacy concerns [98]. Administered subcutaneously, it may result in the following adverse effects: upper respiratory tract infection, diarrhoea, nausea, injection site reaction, cough, back pain, arthralgia, sinusitis, and influenza [97].

#### 3.2.5. Dulaglutide (Trulicity)

Dulaglutide is a 31-mer linear acid peptide (H-HGEGTFTSDVSSYLEEQAAKEFIAWLVKGGG-OH) that belongs to GLP-1 family (Figure 25). It is used alongside diet and exercise to enhance glycemic control in adults diagnosed with type 2 diabetes mellitus. Additionally, it is utilised to lower the risk of major adverse cardiovascular events in adults with type 2 diabetes mellitus who have established cardiovascular disease or multiple cardiovascular risk factors [100]. 

Dulaglutide, a GLP-1 receptor agonist, binds to and activates the GLP-1 receptor, resulting in an augmentation of glucose-dependent insulin synthesis and the in vivo secretion of insulin from pancreatic β cells [100]. Dulaglutide elevates the levels of cAMP, along with engaging other downstream signalling pathways. This process culminates in the glucose-dependent release of insulin, a reduction in glucagon secretion, and a deceleration of gastric emptying [100,101]. After subcutaneous administration, the time to reach the maximum plasma concentration of dulaglutide at a steady state varies from 24 to 72 h, with a median of 48 h. Following once-weekly administration, steady-state plasma dulaglutide concentrations are achieved between 2 and 4 weeks, and the accumulation ratio is approximately 1.56. Importantly, the site of subcutaneous administration (abdomen, upper arm, and thigh) does not have a statistically significant effect on the exposure to dulaglutide [100].

Dulaglutide was developed by Eli Lilly and Company and received FDA approval in 2014 [102,103]. Administered subcutaneously, it has been associated with the following adverse effects: nausea, diarrhoea, vomiting, abdominal pain, and decreased appetite [100]. Dulaglutide exhibited an apparent population mean clearance of 0.142 L/h, with an elimination half-life of around 5 days [100]. Dulaglutide is presumed to undergo degradation into its component amino acids through general protein catabolism pathways [100].

#### 3.2.6. Semaglutide (Ozempic)

Semaglutide is a 31-mer acid peptide that belongs to GLP-1 family. It possesses a Glu-containing moiety appended to its Lys side chain, along with two mini-PEG amino acids (8-amino-3,6-dioxaoctanoic acid, ADO) and a C18 diacid. The inclusion of the alpha-aminoisobutyric acid (Aib) residue contributes to the enhanced stability of the peptide against DDP-IV (Figure 26) [104,105,106]. It is as an adjunct treatment alongside diet and exercise to enhance glycemic control in adults diagnosed with type 2 diabetes mellitus [107].

Semaglutide, as a GLP-1 receptor agonist, binds to and activates the GLP-1 receptor. Similar to the actions of GLP-1, it carries out several effects on glucose regulation through the activation of the GLP-1 receptor [108]. It stimulates the insulin secretion and inhibits the glucagon secretion in a glucose-dependent manner [107]. It exhibits extensive binding to albumin protein, enabling a once-weekly administration [107]. The absolute bioavailability of semaglutide is 89%. The maximum concentration of semaglutide is typically reached 1 to 3 days post-dose. Additionally, similar exposure levels are attained with subcutaneous administration of semaglutide in the abdomen, thigh, or upper arm [107].

Semaglutide was developed by Novo Nordisk and received FDA approval in 2017 [109]. Administered subcutaneously, it may lead to side effects such as nausea, vomiting, diarrhoea, abdominal pain, and constipation [107]. The main route of elimination for semaglutide is through metabolism, involving proteolytic cleavage of the peptide backbone and sequential beta-oxidation of the fatty acid side chain [107].

#### 3.2.7. Tirzepatide (Mounjaro) 

Tirzepatide is a 39-mer peptide amide, functioning as a GLP-1 analogue, and it incorporates a C20 fatty acid diacidic moiety. This structural feature facilitates binding to albumin, thereby extending its half-life [110]. The inclusion of two Aib nonproteinogenic residues contributes to the stability of the peptide by protecting it against peptidase degradation (Figure 27) [92]. It is used as a treatment, in conjunction with diet and exercise, to enhance glycemic control in adults diagnosed with type 2 diabetes mellitus [110].

Tirzepatide acts as an agonist for both GLP-1 and GIP [111]. Tirzepatide selectively binds to both GIP and GLP-1 receptors, activating them. This activation enhances the phases of insulin secretion and reduces glucagon levels in a glucose-dependent manner [110]. After subcutaneous administration, the time to reach maximum plasma concentration of tirzepatide varies from 8 to 72 h. The mean absolute bioavailability of tirzepatide following subcutaneous administration is 80%. Additionally, similar exposure levels are achieved with subcutaneous administration of tirzepatide in the abdomen, thigh, or upper arm. [110].

Tirzepatide was developed by Eli Lilly and Company and received FDA approval in 2022 [112]. Administered subcutaneously, tirzepatide has been associated with various adverse effects, including nausea, diarrhoea, decreased appetite, vomiting, constipation, dyspepsia, and abdominal pain [110]. The apparent population mean clearance of tirzepatide is 0.061 L/h, and it has an elimination half-life of approximately 5 days. This prolonged half-life allows for once-weekly dosing [110]. Tirzepatide undergoes metabolism through proteolytic cleavage of the peptide backbone, beta-oxidation of the C20 fatty diacid moiety, and amide hydrolysis [110].

## 4. Growth-Hormone-Releasing Hormone (GHRH) Analogues

Peptides have found a role in mimicking ghrelin, a 28-mer peptide synthesized in the stomach. Ghrelin stimulates the secretion of growth hormone (GH) by interacting with a specific receptor known as the GH secretagogue receptor 1a (GHS-R1a) [113]. Activating this receptor through agonists proves beneficial for various pharmacological applications, including addressing issues such as growth retardation, gastrointestinal dysfunction, and impaired body composition (Figure 28) [113]. Table 3 shows four of the FDA-approved GHRH analogues.

### 4.1. Sermorelin (Geref)

Sermorelin is a 29-mer peptide amide (H-YADAXFXNSYRKVLGQLSARKLLQDXMSR-NH_2_) and is a synthetic analogue of GHRH (Figure 29) [114]. It is used to reduce the excess abdominal fat in adult patients infected with the human immunodeficiency virus (HIV) and experiencing lipodystrophy [114].

Sermorelin binds and activates the receptor for GHRH, mimicking the natural GRF by effectively stimulating the secretion of GH [113,114]. The subcutaneous administration of a 2 mg dose of EGRIFTA (1 mg/vial formulation) resulted in an absolute bioavailability of tesamorelin of less than 4% in healthy adult subjects [114].

Sermorelin was developed by Serono Laboratories and approved by the FDA in 1991 [115]. Administered subcutaneously, it may lead to the following adverse effects: arthralgia, erythema at the injection site, pruritus at the injection site, pain in the extremities, peripheral oedema, and myalgia [114]. After the administration of a single 1.4 mg subcutaneous dose, sermorelin exhibited a mean elimination half-life of 8 min in healthy subjects [114].

### 4.2. Mecasermin (Increlex)

Mecasermin is a 70-mer acid peptide, representing human insulin-like growth factor-1 (rhIGF-1). It features three disulfide bridges linking Cys6 to Cys48, Cys18 to Cys61, and Cys47 to Cys52, maintaining the identical sequence as endogenous human IGF-1 (Figure 30) [116,117]. The presence of disulfide bridges is essential, as the non-matching forms lack affinity for IGF-1 [118]. Moreover, disulfide bridges ease the biotransformation process, particularly with endogenous thiols in the blood, and this is regarded as a primary metabolic pathway [118]. Mecasermin serves as a prolonged therapeutic option for addressing growth failure in children diagnosed with severe primary insulin-like growth factor-1 deficiency (primary IGFD) or those with a GH gene deletion who have developed neutralizing antibodies to GH [119,120].

Within target tissues, mecasermin binds to and activates the IGF-1 receptor, a receptor homologous to the insulin receptor. This activation initiates intracellular signalling, subsequently stimulating various processes that contribute to statural growth. These processes include mitogenesis in multiple tissue types, growth and division of chondrocytes along cartilage growth plates, and an augmentation in organ growth [116,121]. The reported bioavailability after subcutaneous administration in healthy subjects is close to 100%, although there are no available data for subjects with primary insulin-like growth factor-1 deficiency [116].

Mecasermin was developed by Tercica and was approved by the FDA in 2005 [119]. Administered subcutaneously, it may lead to the following adverse effects: low blood sugar, enlarged tonsils, or an allergic reaction [116,121]. The estimated mean terminal half-life after a single subcutaneous administration of 0.12 mg/kg of mecasermin in paediatric subjects with severe primary IGFD is 5.8 h [116]. It has been demonstrated that both the liver and the kidney play a role in metabolizing mecasermin [116].

### 4.3. Tesamorelin (Egrifta)

Tesamorelin is classified as a synthetic analogue of GHRH. It incorporates a hexenoyl moiety, characterised by a C6 chain with a double bond at position 3, linked to the *N*-terminal Tyr of the peptide (Figure 31). The inclusion of a hexenoyl moiety led to increased stability in serum when compared to natural human GHRH. Tesamorelin is ustilised to mitigate the surplus abdominal fat in patients infected with the human immunodeficiency virus (HIV) and experiencing lipodystrophy [122].

Tesamorelin binds to and activates the receptors of the GHRH [123], enhancing the release of GH. This, in turn, interacts with various receptors on different cells, including chondrocytes, osteoblasts, myocytes, hepatocytes, and adipocytes, leading to a myriad of pharmacodynamic effects [122]. The absolute bioavailability of tesamorelin was found to be less than 4% in healthy adult subjects following subcutaneous administration of a 2 mg dose [122].

Tesamorelin was developed by Kendle International and was approved by the FDA in 2010 [124]. Administered subcutaneously, it may result in the following adverse effects: arthralgia, erythema at the injection site, pruritus at the injection site, pain in the extremities, peripheral oedema, and myalgia [122]. The mean elimination half-life of tesamorelin was 26 min in healthy subjects and 38 min in HIV-infected patients after subcutaneous administration for 14 consecutive days [122].

### 4.4. Macimorelin (Macrilen)

Macimorelin is a small peptide that has only three residues: Aib, D-Trp, and a gem-diamino moiety that mimics D-Trp (Figure 32). The incorporation of D-amino acids and Aib imparts stability to the peptide, safeguarding it against proteolytic degradation and DPP-IV activity, respectively. It is used for the diagnosis of adult GH deficiency [125].

Macimorelin acts as a GH secretagogue receptor agonist, stimulating the ghrelin receptor (GHS-R1a) and triggering the release of GH [125]. After administering 0.5 mg of macimorelin per kilogram of body weight orally to patients with adult GH deficiency (AGHD) who had fasted for at least 8 h, the maximum plasma concentrations (C_max_) were observed between 0.5 and 1.5 h [125].

The discovery of macimorelin originated from the work of Fehrentz and Martinez at the University of Montpellier [126]. Aeterna Zentaris GmbH led the development process, resulting in FDA approval in 2017 [127]. It is administered orally, and may cause several adverse effects, including dysgeusia, dizziness, headache, fatigue, nausea, hunger, diarrhoea, upper respiratory tract infection, feeling hot, hyperhidrosis, nasopharyngitis, and sinus bradycardia [125]. Macimorelin exhibited a mean terminal half-life of 4.1 h after the administration of a single oral dose of 0.5 mg macimorelin per kilogram of body weight in healthy subjects [125]. Results from an in vitro study using human liver microsomes indicated that CYP3A4 is the primary enzyme responsible for metabolizing macimorelin [125].

## 5. Cholecystokinin Analogues (CCK)

Cholecystokinin (CCK) is a brain–gut peptide released by enteroendocrine cells primarily located in the upper small intestine, specifically the duodenum and jejunum. Its secretion is triggered by the presence of protein, fat, and nutrients [128]. It consists of four conserved amino acids at its *C*-terminal (H-WMDF-NH2), with a sulphated tyrosine in position 7 [129]. Various receptors participate in nutrient-induced CCK release, resulting in a common outcome: an elevation of intracellular calcium levels. This increase subsequently triggers muscle contraction and the release of CCK [130]. Therefore, CCK plays a vital role in regulating food intake. The absence of its signal delays the feeling of satiety, leading to the consumption of larger meals (Figure 33) [131].

Gibbs and his colleagues demonstrated the capacity of exogenous CCK to suppress food intake in rats [128]. Consequently, therapeutic peptide analogues were developed to serve as exogenous sources of CCK.

### Sincalide (Kinevac)

Sincalide is an 8-mer peptide amide and a CCK analogue (Figure 34) [132]. It is used to induce gallbladder contraction, promote pancreatic secretion, and hasten the passage of a barium meal through the small bowel. This action reduces the duration and extent of radiation exposure associated with fluoroscopy and X-ray examinations of the intestinal tract [132].

Sincalide induces gallbladder contraction and diminishes its size. Additionally, it promotes pancreatic secretion and intestinal motility, leading to pyloric contraction and delayed gastric emptying [132]. Upon administering a bolus injection of 0.02 mcg/kg of sincalide intravenously, the gallbladder exhibited maximum contraction within 5 to 15 min. Sincalide achieved a reduction in gallbladder radiographic size by at least 40%, which is generally considered a satisfactory contraction [132].

Sincalide was developed by Fresenius Kabi and was approved by the FDA in 1976 [133]. It is administered intravenously and is associated with adverse effects such as abdominal discomfort or pain, as well as nausea [132].

## 6. Adrenocorticotropic Hormone (ACTH) and Analogues

Adrenocorticotropic hormone (ACTH) is a 39-mer acid peptide (H-SYSMEHFRWGKPVGKKRRPVKVYPNGAEDESAEAFPLEF-OH), classified as a tropic hormone. It is synthesized by the anterior pituitary gland and is regulated by the hypothalamic-pituitary axis [134] in response to the corticotropin-releasing hormone (CRH) from the hypothalamus, as well as the hypothalamic hormone arginine vasopressin [135]. ACTH regulates the production of cortisol and androgen. Its receptors are G protein-coupled receptors, which act by stimulating adenyl cyclase. This stimulation increases cAMP levels, thereby activating PKA (Figure 35) [135]. The FDA has approved two ACTH peptide-based analogues (Table 4).

### 6.1. Corticorelin (Acthrel)

Corticorelin is a 41-mer peptide amide with a sequence identical to that of ovine corticotropin-releasing hormone (oCRH) (H-SQEPPISLDLTFHLLREVLETTKADQLAQQAHSNRKLLDIA-NH_2_) (Figure 36). It is utilised as a diagnostic agent to assess the status of the pituitary–adrenal axis, aiding in distinguishing between a pituitary source and an ectopic source of excessive ACTH secretion in ACTH-dependent Cushing’s syndrome [136].

Corticorelin is a potent stimulator of ACTH release from the anterior pituitary gland. ACTH, in turn, stimulates cortisol production from the adrenal cortex. Following the injection of corticorelin doses of ≥0.3 mcg/kg, normal subjects experienced a rise in plasma ACTH levels within 2 min, reaching peak levels at 10–15 min. Simultaneously, plasma cortisol levels increased within 10 min, peaking at 30 to 60 min. With an increase in the corticorelin dose, the elevations in plasma ACTH and cortisol displayed a more prolonged pattern, exhibiting a biphasic response with a second, lower peak observed at 2–3 h post-injection. [136].

Corticorelin was developed by Ferring Pharmaceuticals and received FDA approval in 1996 [137]. In 2020, the manufacturing company discontinued corticorelin due to sourcing issues. When administered intravenously, it may lead to adverse effects including flushing of the face, neck, and upper chest; dyspnea; wheezing; urticaria; angioedema; tachycardia; hypotension; and sensations of “chest compression” or tightness [136].

### 6.2. Corticotropin (Cosyntropin)

Corticotropin is a synthetic 24-mer peptide derived from the first 24 amino acid residues of the 39-amino acid sequence of ACTH, (H-SYSMEHFRWGKPVGKKRRPVKVYP-NH_2_) (Figure 37). Studies have demonstrated that the activity of ACTH resides in its *N*-terminal portion, specifically within a minimum of 20 amino acid residues [32]. It is used in the diagnosis of patients suspected to have adrenocortical insufficiency [32].

Corticotropin binds to a specific receptor on the plasma membrane of the adrenal cells. This binding initiates a cascade of reactions involved in the synthesis of adrenal steroids, such as cortisol, cortisone, weak androgenic substances, and a limited amount of aldosterone. One of the primary effects is an increase in the quantity of the substrate within the mitochondria, facilitating steroid synthesis [32].

Corticotropin was developed by Sandoz and approved by the FDA in 2008 [33]. It is administered intravenously, and may lead to adverse effects such as bradycardia, tachycardia, hypertension, peripheral oedema, and rash [32].

## 7. α-Melanocyte Stimulating Hormone (α-MSH) Analogues

α-Melanocyte stimulating hormone (α-MSH) is an endogenous hormone and a neuropeptide belonging to the melanocortin family. It is an acetylated 13-mer peptide amide with the sequence Ac-SYSMEHFRWGKPV-NH_2_ [138] and is derived from proopiomelanocortin (POMC), a shared precursor protein for all melanocortin peptides, including α-MSH and ACTH [139]. Activation of the POMC neurons has been demonstrated to initiate the production and release of α-MSH from the POMC axon terminals [138,139]. There are five melanocortin receptors (MCRs), numbered MC1R through MC5R [138]. All of them, with the exception of MC2R, are found in the brain and are activated by either α-MSH or ACTH. MC2R is specifically present in adrenal cells and is activated exclusively by ACTH (Figure 38) [139]. MSH binding sites seem to be widely distributed throughout the central nervous system. Therefore, MSH has diverse effects on functions such as learning and memory, fever suppression, peripheral nerve regeneration, inflammatory and immune responses, and sexual behaviour, among others [138,139]. Synthetic α-MSH peptides can elicit a therapeutic effect on those receptors comparable to that of their endogenous counterparts [140]. Table 5 shows the FDA approved peptide-based α-MSH analogues. 

### 7.1. Afamelanotide (Scenesse)

Afamelanotide is a 13-mer peptide amide (H-SYSnorLEHFRWGKPV-NH_2_) whose sequence closely resembles that of α-MSH, with the exception of two amino acids. Specifically, in afamelanotide, Met4, and L-Phe7 are substituted with norleucine (NLe4) and D-Phe7, respectively, to achieve its structure (Figure 39). Those modifications are critical for enhancing the stability of afamelanotide against enzymatic degradation, consequently prolonging its plasma half-life [141]. It acts as an MC1R agonist and is utilised to enhance pain-free light exposure in adult patients with a history of phototoxic reactions from erythropoietic protoporphyria (EPP) [142].

Afamelanotide primarily binds to MC1R and promotes the synthesis of eumelanin, which provides essential photoprotection [142,143,144]. Significant variability was observed in the plasma concentrations of afamelanotide, with the last measurable concentration for most subjects (9 out of 12) recorded at 96 h post-dose. The mean maximum concentration (C_max_) was 3.7 ± 1.3 ng/mL, and the area under the concentration–time curve (AUC_0-inf_) was 138.9 ± 42.6 h·ng/mL [142].

Afamelanotide was developed by Clinuvel and was approved by the FDA in 2019 [145]. It is worth noting that the development story of afamelanotide began earlier, originating at the University of Arizona in the 1980s [146]. It is administered subcutaneously, and may lead to adverse effects, including implant site reactions, nausea, oropharyngeal pain, cough, fatigue, dizziness, skin hyperpigmentation, somnolence, melanocytic nevus, respiratory tract infection, non-acute porphyria, and skin irritation [142]. When administered subcutaneously through a controlled-release implant, afamelanotide exhibits an apparent half-life of approximately 15 h [142]. Hydrolysis may be a potential metabolic pathway for afamelanotide. However, the complete metabolic profile of afamelanotide has not yet been thoroughly characterised [142].

### 7.2. Bremelanotide (Vyleesi)

Bremelanotide is an analogue of α-MSH, distinguished by its homodetic cyclic peptide structure, Ac-Nle-cyclo[DHfRWK]-OH. The cyclic structure involves an amide bond between the ß carboxylic acid of Asp and the ε amino of the Lys, which serves as the *C*-terminal residue (Figure 40) [147]. The cyclic structure and incorporation of D-amino acids contribute to the stability of the peptide. Furthermore, acetylation exerts influence on various protein functions, such as enzymatic activity, stability, DNA binding, protein–protein interaction, and recognition by peptide receptors. This modification occurs across a wide range of proteins, showcasing its diverse and widespread impact [148].

Bremelanotide is an MC1R agonist used in the treatment of premenopausal women with acquired, generalized hypoactive sexual desire disorder (HSDD). This disorder is characterised by low sexual desire causing significant distress or interpersonal difficulty and is not attributed to co-existing medical or psychiatric conditions, relationship problems, or the effects of medication or substance use [149].

Bremelanotide non-selectively activates various receptor subtypes, with the most pertinent binding occurring with MC1R and MC4R at the therapeutic dose [149]. Neurons expressing MC4R are distributed throughout various regions of the CNS. MC4R is predominantly expressed in the medial preoptic area (mPOA) of the hypothalamus in the brain and is important for female sexual function [150]. However, the exact mechanism by which activation of MC4R enhances HSDD in women is not yet fully understood [149]. Additionally, bremelanotide binds to and activates MC1R, triggering melanocytes to generate melanin, subsequently enhancing pigmentation [149].

Bremelanotide exhibits a median time to maximum plasma concentration (T_max_) of approximately 1.0 h, with a range of 0.5 to 1.0 h. The absolute bioavailability of bremelanotide after subcutaneous administration is approximately 100%. Furthermore, the choice of a subcutaneous administration site, whether in the abdomen or thigh, does not significantly impact the systemic exposure to bremelanotide [149].

Bremelanotide has undergone various stages of development, with contributions from both the Arizona Cancer Research Center and Palatin Technologies [151]. However, Amag Pharmaceuticals obtained FDA approval for bremelanotide in 2019 [152]. Administered subcutaneously, it has exhibited some adverse effects, including nausea, flushing, injection site reactions, headache, and vomiting [149]. Bremelanotide has an average terminal half-life of about 2.7 h, ranging from 1.9 to 4.0 h, and the mean clearance (CL/F) is 6.5 ± 1.0 L/h [149]. The primary metabolic route for bremelanotide comprises successive hydrolytic processes of the amide bond within the cyclic peptide [149].

### 7.3. Setmelanotide (Imcivree)

Setmelanotide is an 8-mer cyclic peptide amide with the sequence Ac-RCaHfRWC-NH_2_. It features a disulfide bridge between Cys2 and Cys8, contributing to its structural stability. Additionally, the incorporation of two D-amino acid residues enhances its resistance to enzymatic degradation. This structural design aims to improve the peptide’s durability in biological systems, extending its therapeutic effectiveness (Figure 41). Setmelantide is an MC4R agonist used in the treatment of chronic weight management. It is specifically indicated for adult and pediatric patients aged 6 years and older with obesity caused by POMC, proprotein convertase subtilisin/kexin type 1 (PCSK1), or leptin receptor (LEPR) deficiency. Confirmation of these genetic conditions is determined through genetic testing, which should show variants in POMC, PCSK1, or LEPR genes that are interpreted as pathogenic, likely pathogenic, or of uncertain significance (VUS). Setmelanotide aims to address obesity in individuals with these specific genetic conditions [153]. Setmelanotide is metabolised through catabolic pathways, leading to the formation of small peptides [153].

Setmelanotide functions by binding preferentially to MC4R and activating it. This activation re-establishes the pathway activity associated with MC4R, leading to a reduction in hunger and promoting weight loss. The mechanism involves a decrease in caloric intake and an increase in the expenditure of energy, ultimately contributing to the management of obesity in individuals with specific genetic conditions such as POMC, PCSK1, or LEPR deficiencies [153,154,155,156,157]. After the administration of setmelanotide via subcutaneous injection, the peak plasma concentrations of setmelanotide were observed at a median t_max_ of 8 h post-dosing [153].

Setmelanotide was developed by Rhythm Pharmaceuticals and received FDA approval in 2020 [158]. It is administered subcutaneously and has the following adverse effects: injection site reactions, skin hyperpigmentation, nausea, headache, diarrhoea, abdominal pain, back pain, fatigue, vomiting, depression, upper respiratory tract infection, and spontaneous penile erection [153]. The effective elimination half-life of setmelanotide is approximately 11 h, and the estimated total apparent steady-state clearance following subcutaneous administration of setmelanotide at a daily dose of 3 mg is 4.86 L/h, as determined by the population pharmacokinetic model [153].

## 8. Conclusions

Therapeutic peptides have expanded beyond their natural origins. The medicinal chemistry discipline has enabled the diversification of the structures of various medications, including peptides, making it possible to address previously unmet medical needs. Since the discovery of insulin, peptides continue to capture a substantial percentage of FDA approvals each year. 

Peptides encounter significant challenges related to their stability against enzymatic attacks. Consequently, chemical modifications are considered essential to uphold their therapeutic efficacy [1]. While peptides were essentially designed based on their natural origin, chemical modifications played a central role in boosting their binding to albumin, hence prolonging their half-lives and reducing the frequency of required administration [79]. On the other hand, there remains an absence of innovative approaches aimed at improving peptide stability and thereby enabling a broader utilisation of orally administered peptides. Among approximately 102 FDA-approved peptides, only 11 can be administered orally. This highlights the imperative to explore novel strategies for developing groundbreaking peptide analogues. Undoubtedly, the witnessed advancement in peptide therapeutics can be ascribed to significant advancement in synthetic technologies. The significance of advancing synthetic strategies cannot be overstated, as it enables the synthesis of novel peptide structures, pushing the boundaries of exploration in the field. Additionally, advancing the understanding of the target receptors’ nature and maximising engagement efficiency are crucial elements for a successful drug development endeavour.

The FDA granted approval to a total of 31 peptide-based drugs between 2016 and 2023 [159]. I foresee a significant rise in approvals in the upcoming years, surpassing the threshold of 60 new approvals by 2030, barring any unforeseen circumstances similar to the downtime experienced during the COVID-19 pandemic.

## Figures and Tables

**Figure 1 biomolecules-14-00264-f001:**
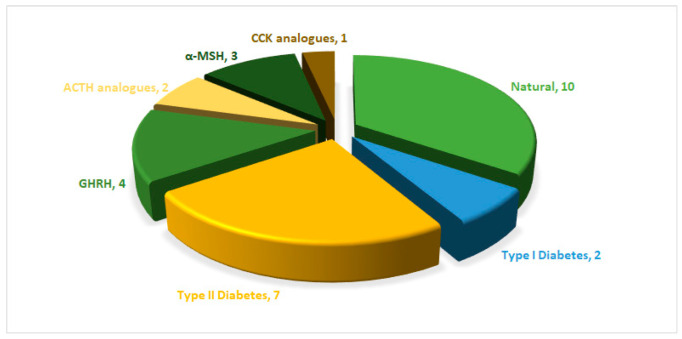
FDA approved natural and engineered peptide analogues (1923–2023). ACTH, adrenocorticotropic hormone; CCK, cholecystokinin; GHRH, growth-hormone-releasing hormone; α-MSH, α-melanocyte-stimulating hormone.

**Figure 2 biomolecules-14-00264-f002:**
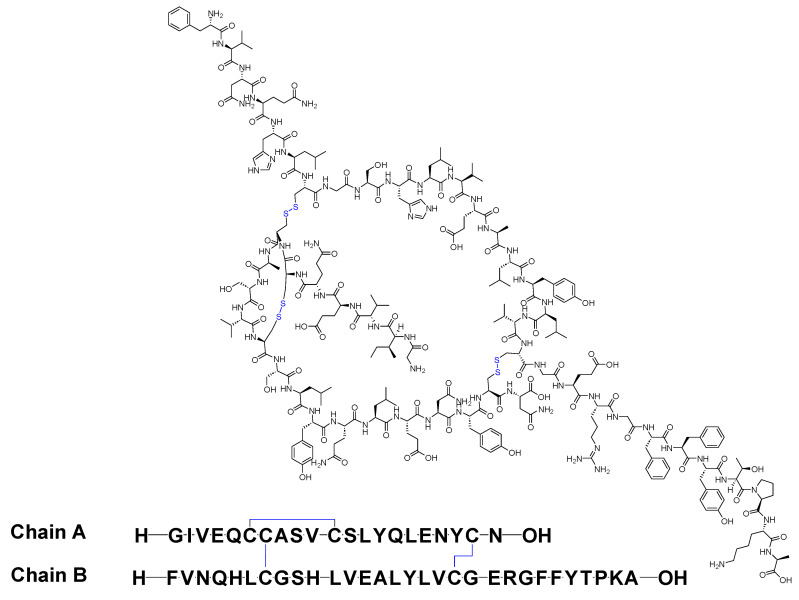
Chemical structure of insulin. Blue: disulfide bridges.

**Figure 3 biomolecules-14-00264-f003:**
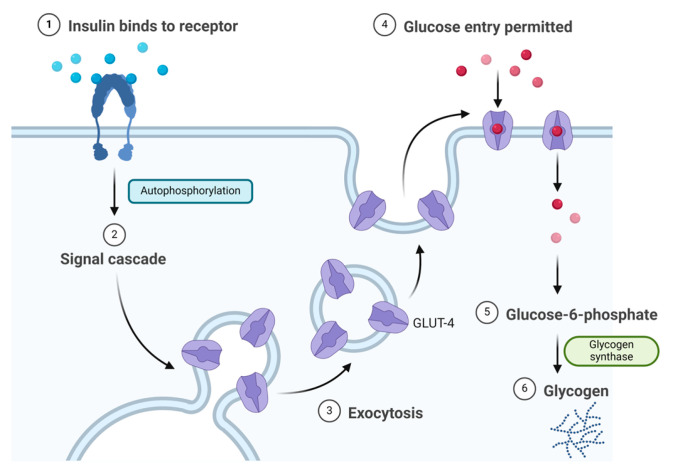
Insulin mechanism of action.

**Figure 4 biomolecules-14-00264-f004:**
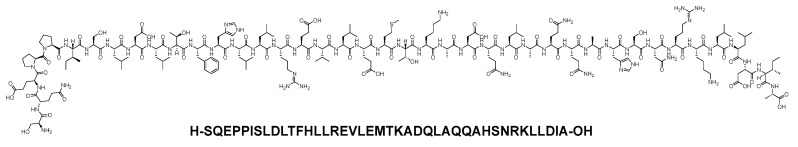
Chemical structure of corticotropin (H.P. Acthar Gel).

**Figure 5 biomolecules-14-00264-f005:**
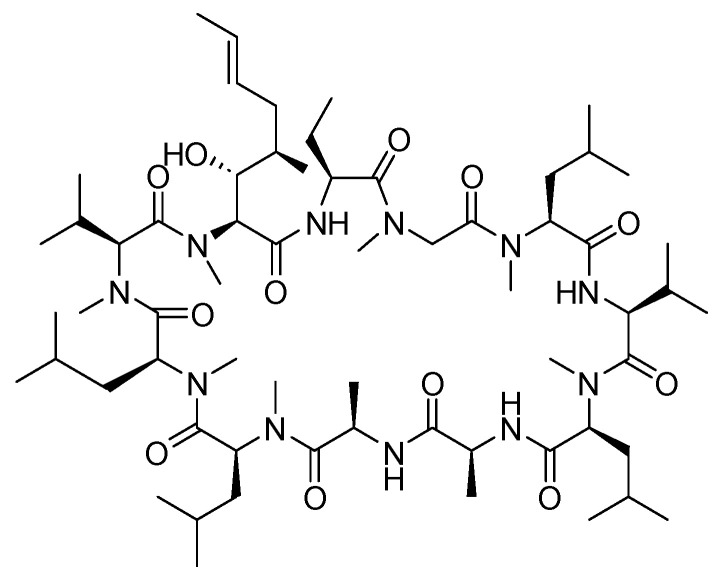
Chemical structure of cyclosporine.

**Figure 6 biomolecules-14-00264-f006:**
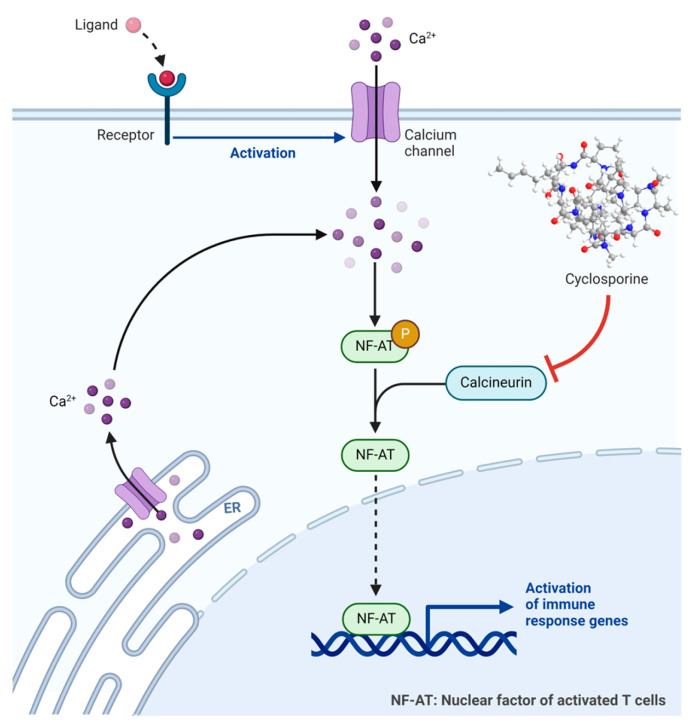
Cyclosporin mechanism of action. NF-AT, nuclear factor of activated T cells.

**Figure 7 biomolecules-14-00264-f007:**
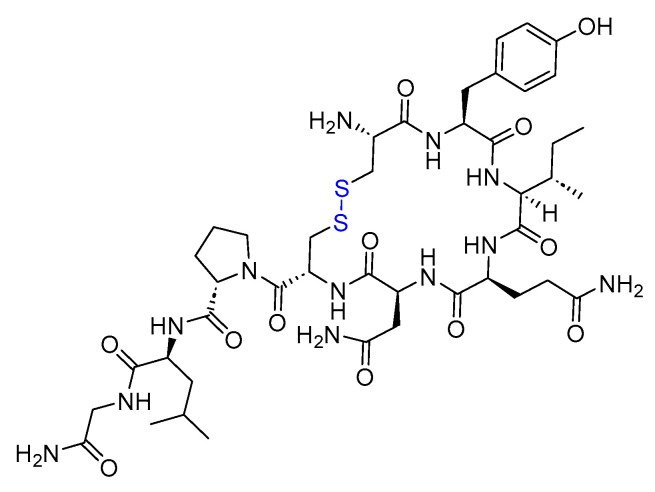
Chemical structure of oxytocin. Blue: disulfide bridge.

**Figure 8 biomolecules-14-00264-f008:**
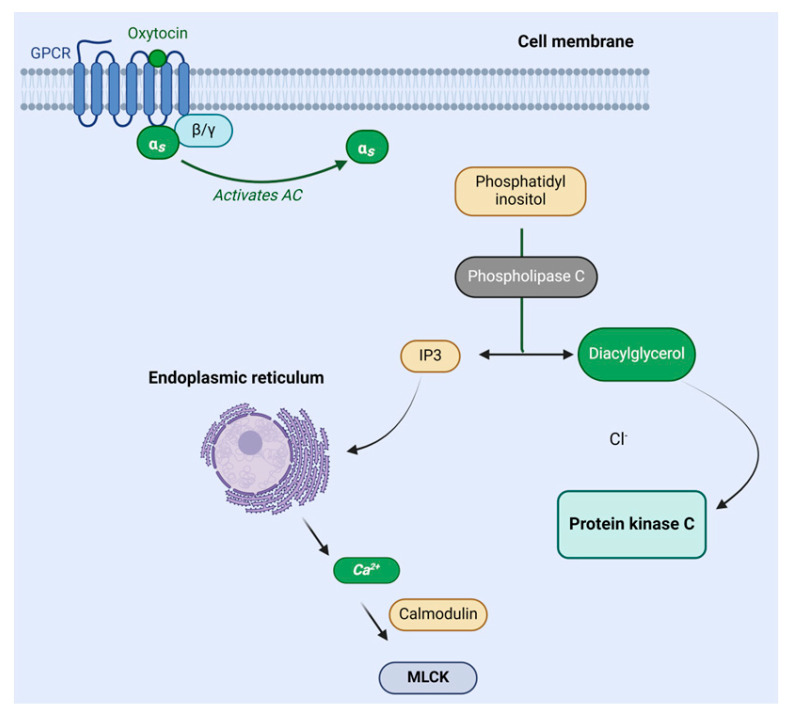
Oxytocin mechanism of action. Adapted from Ref. [51]. IP3, inositol triphosphate; MLCK, myosin light chain kinase enzyme.

**Figure 9 biomolecules-14-00264-f009:**
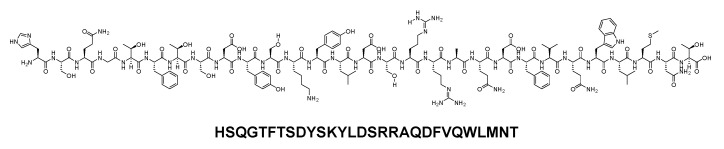
Chemical structure of glucagon.

**Figure 10 biomolecules-14-00264-f010:**
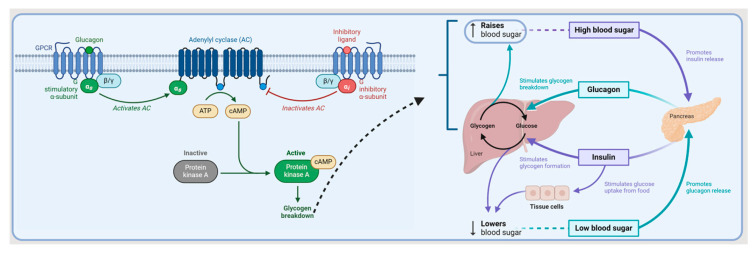
Glucagon mechanism of action. ATP, adenosine triphosphate; cAMP, cyclic adenosine monophosphate.

**Figure 11 biomolecules-14-00264-f011:**
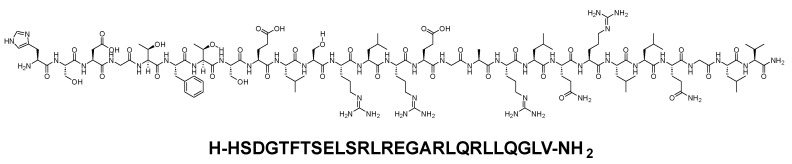
Chemical structure of secretin.

**Figure 12 biomolecules-14-00264-f012:**
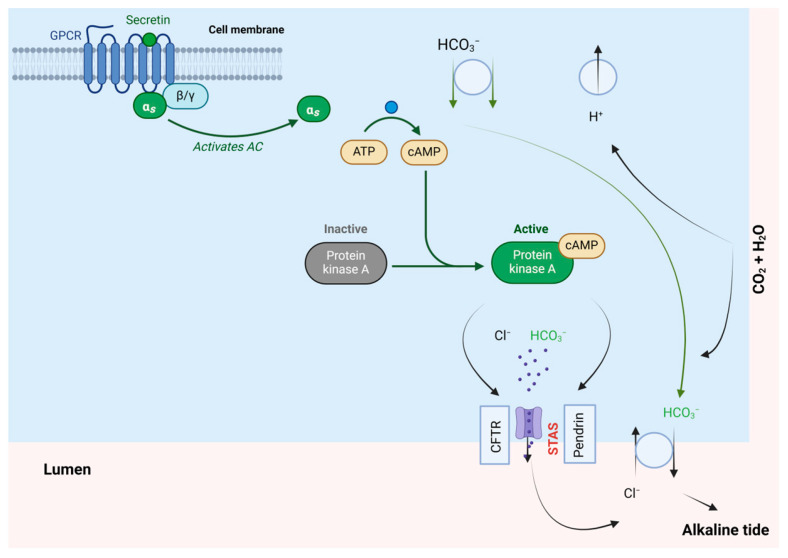
Secretin mechanism of action. ATP, adenosine triphosphate; cAMP, cyclic adenosine monophosphate; CFTR, cystic fibrosis transmembrane conductance regulator; STAS, sulphate transporter and anti-sigma factor antagonist. Adapted from Ref. [59].

**Figure 13 biomolecules-14-00264-f013:**
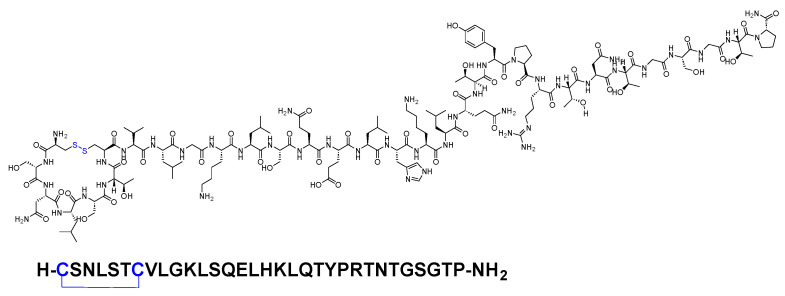
Chemical structure of calcitonin. Blue: disulfide bridge.

**Figure 14 biomolecules-14-00264-f014:**
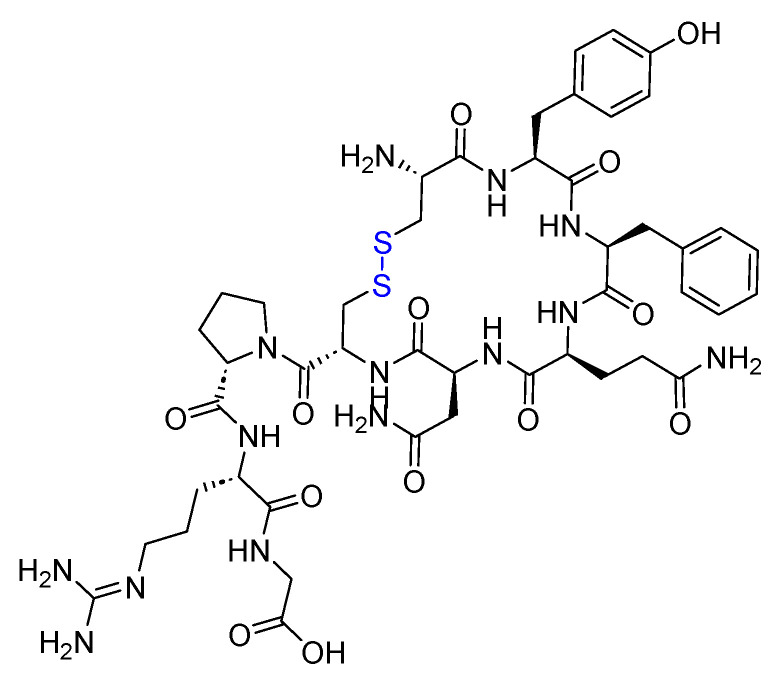
Chemical structure of vasopressin. Blue: disulfide bridge.

**Figure 15 biomolecules-14-00264-f015:**
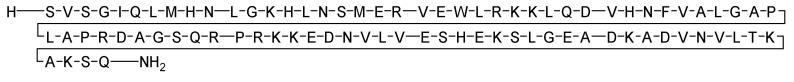
Chemical structure of parathyroid hormone (PTH).

**Figure 16 biomolecules-14-00264-f016:**
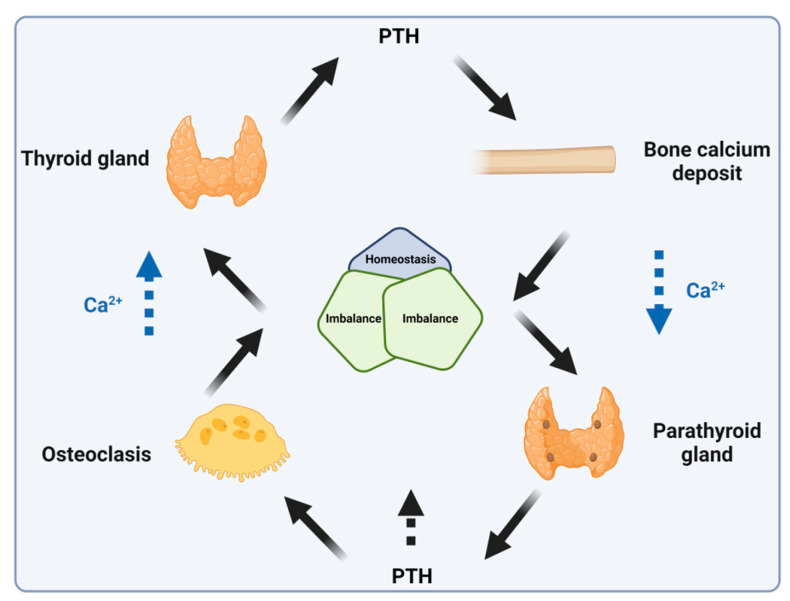
Parathyroid hormone (PTH) mechanism of action. PTH, parathyroid hormone.

**Figure 17 biomolecules-14-00264-f017:**
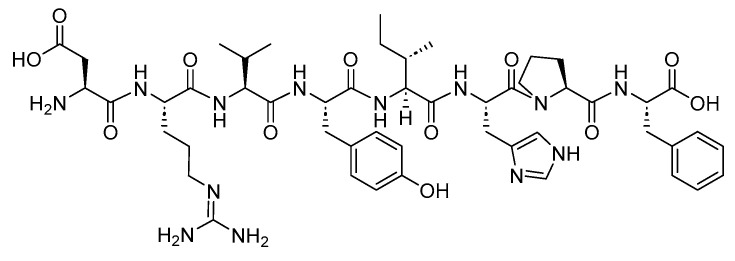
Chemical structure of angiotensin II.

**Figure 18 biomolecules-14-00264-f018:**
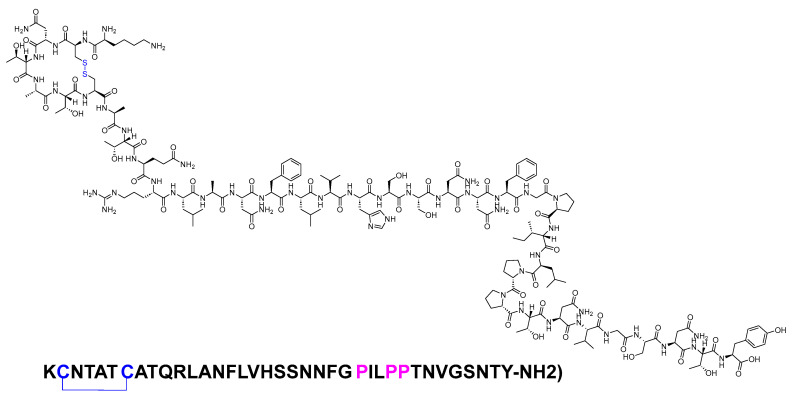
Chemical structure of pramlintide. Blue: disulfide bridge. Pink: positions that are different from those of natural amylin.

**Figure 19 biomolecules-14-00264-f019:**
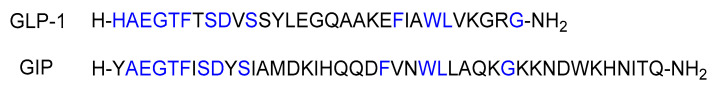
Sequences of GLP-1 and GIP. Blue represents the matching amino acid residues.

**Figure 20 biomolecules-14-00264-f020:**
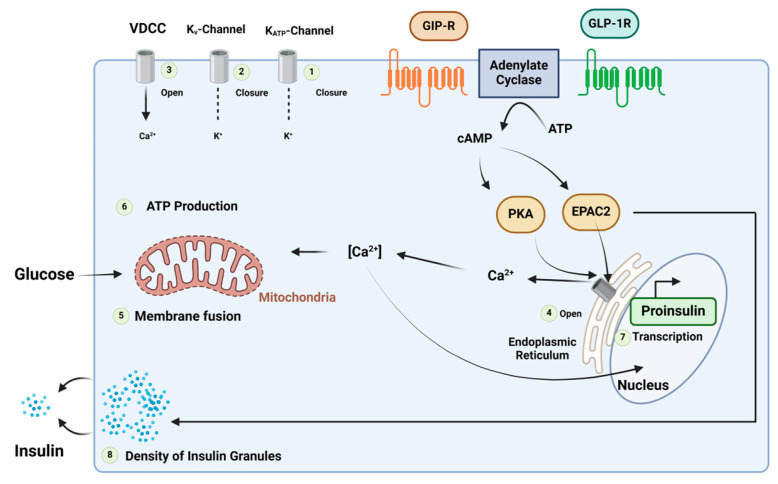
Mechanism of the insulinotropic effects of glucose-dependent insulinotropic polypeptide (GIP) and glucagon-like peptide (GLP)-1. Adapted from Ref. [85].

**Figure 21 biomolecules-14-00264-f021:**
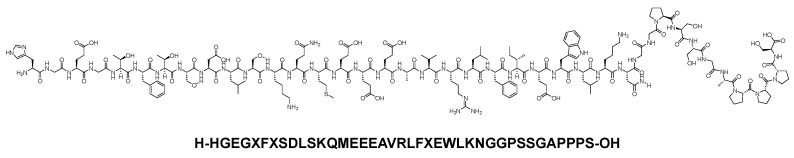
Chemical structure of exenatide.

**Figure 22 biomolecules-14-00264-f022:**
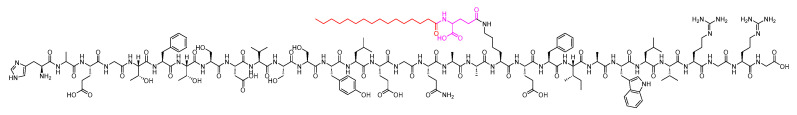
Chemical structure of liraglutide. Black: peptide backbone; red: hexadecanoyl; pink: Glu.

**Figure 23 biomolecules-14-00264-f023:**
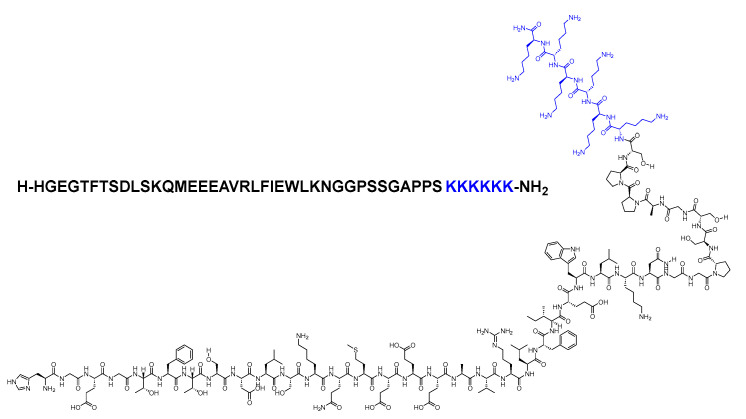
Chemical structure of lixisenatide. Blue: difference from exenatide.

**Figure 24 biomolecules-14-00264-f024:**
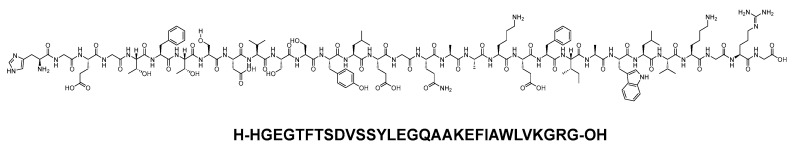
Chemical structure of abiglutide.

**Figure 25 biomolecules-14-00264-f025:**
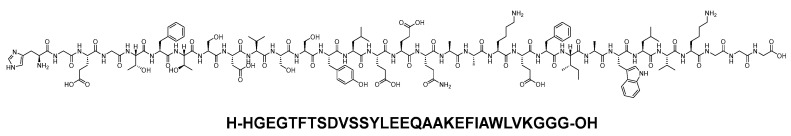
Chemical structure of dulaglutide.

**Figure 26 biomolecules-14-00264-f026:**
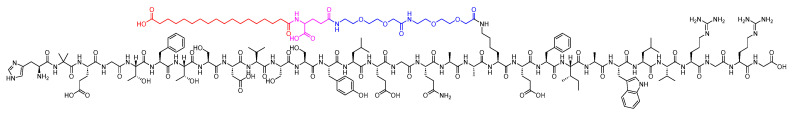
Chemical structure of semaglutide. Black: peptide backbone; red: 17-carboxyheptadecanoyl (C18 diacid); pink: Glu; blue: 8-amino-3,6-dioxaoctanoic acid (ADO).

**Figure 27 biomolecules-14-00264-f027:**
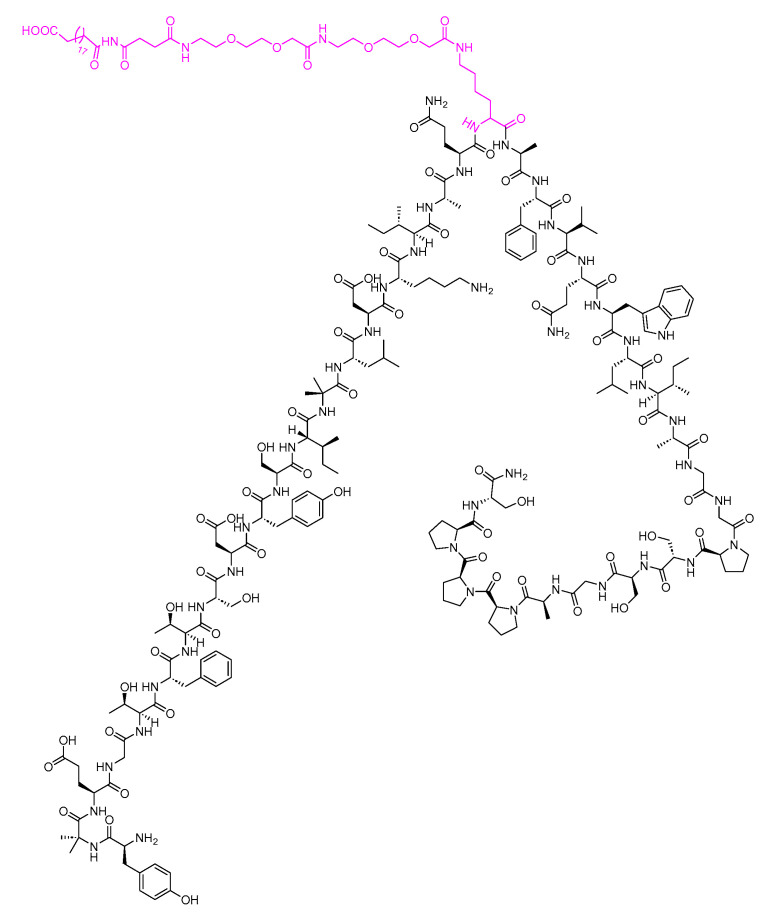
Chemical structure of tirzepatide. Pink: C20 fatty acid diacidic moiety.

**Figure 28 biomolecules-14-00264-f028:**
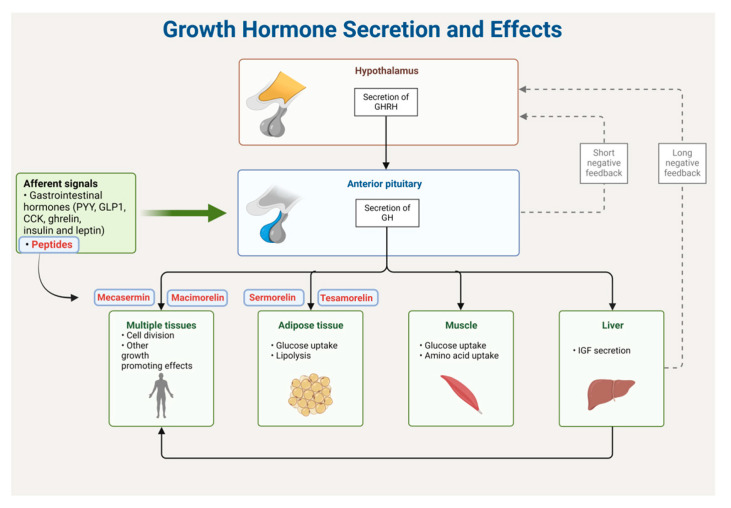
Growth-hormone-releasing hormone (GHRH) mechanism of action. GH, growth hormone.

**Figure 29 biomolecules-14-00264-f029:**
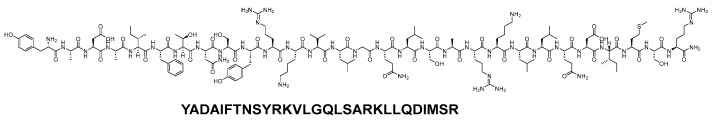
Chemical structure of sermorelin.

**Figure 30 biomolecules-14-00264-f030:**
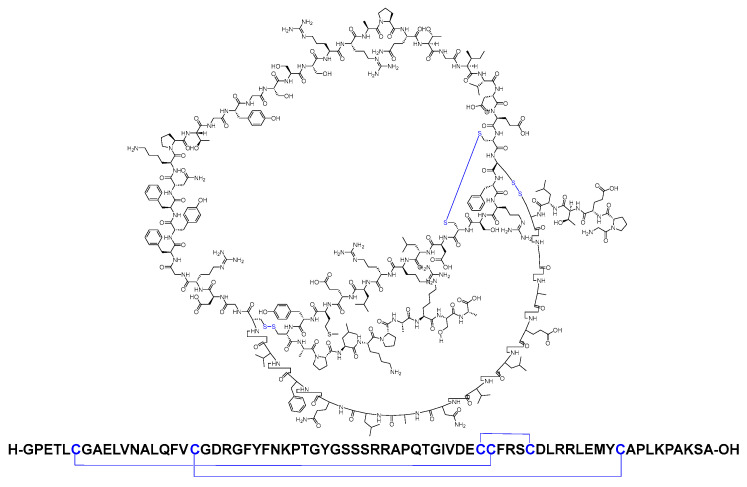
Chemical structure of mecasermin. Blue: disulfide bridge.

**Figure 31 biomolecules-14-00264-f031:**
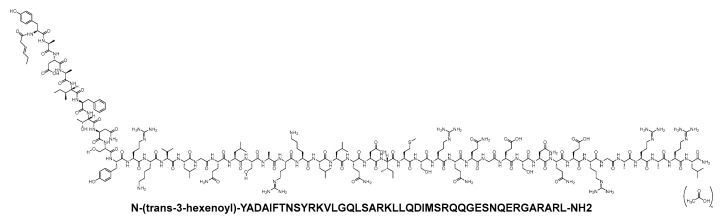
Chemical structure of tesamorelin acetate.

**Figure 32 biomolecules-14-00264-f032:**
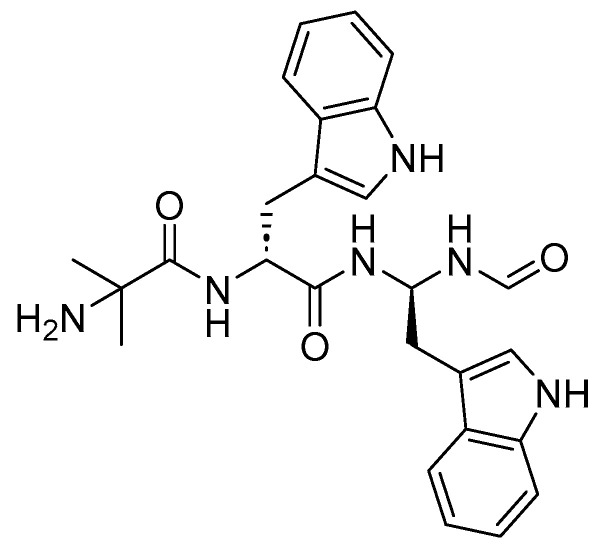
Chemical structure of macimorelin.

**Figure 33 biomolecules-14-00264-f033:**
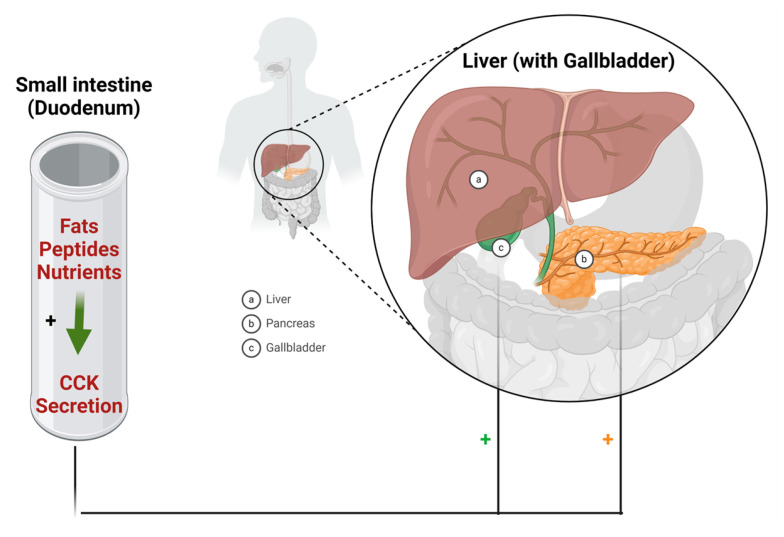
Cholecystokinin (CCK) mechanism of action.

**Figure 34 biomolecules-14-00264-f034:**
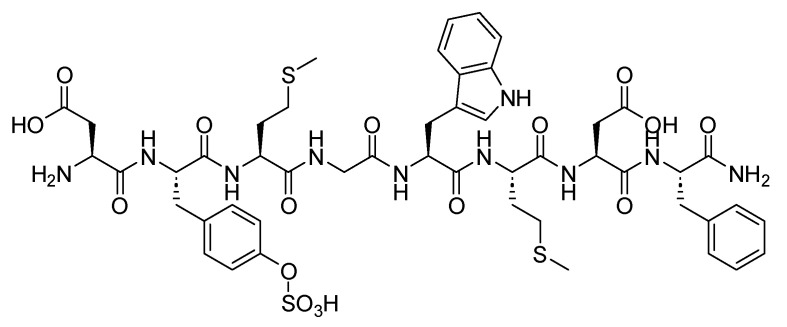
Chemical structure of sincalide.

**Figure 35 biomolecules-14-00264-f035:**
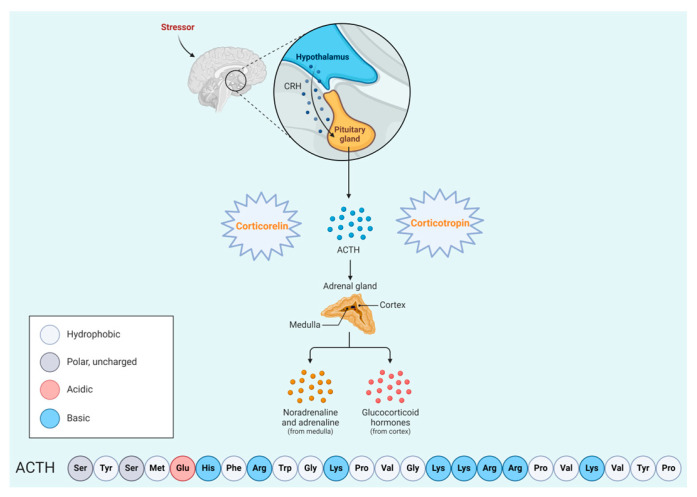
Adrenocorticotropic hormone (ACTH) mechanism of action. The ACTH sequence shown is the effective *N*-terminal portion.

**Figure 36 biomolecules-14-00264-f036:**
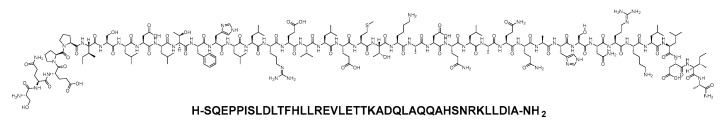
Chemical structure of corticorelin.

**Figure 37 biomolecules-14-00264-f037:**
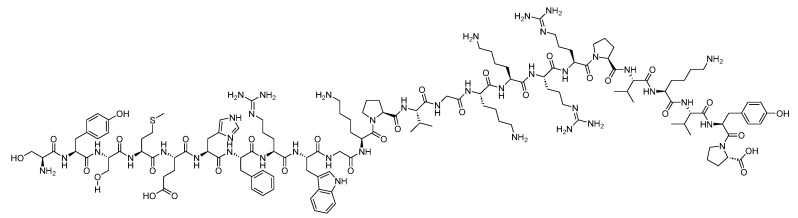
Chemical structure of corticotropin (Cosyntropin).

**Figure 38 biomolecules-14-00264-f038:**
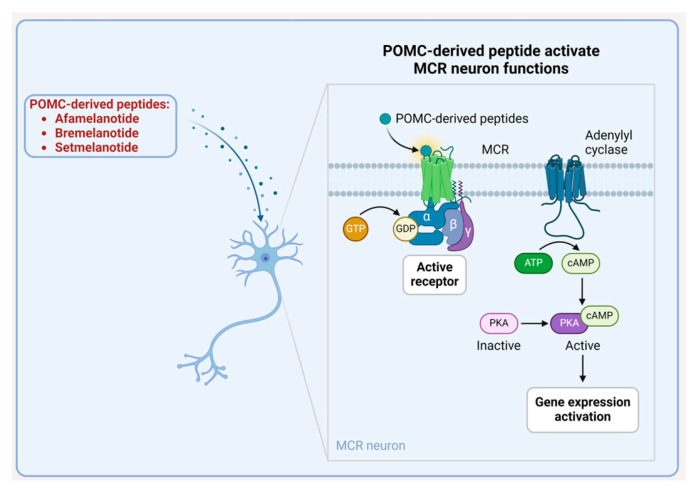
α-Melanocyte stimulating hormone (α-MSH) mechanism of action. MCR, melanocortin receptor; POMC, proopiomelanocortin.

**Figure 39 biomolecules-14-00264-f039:**
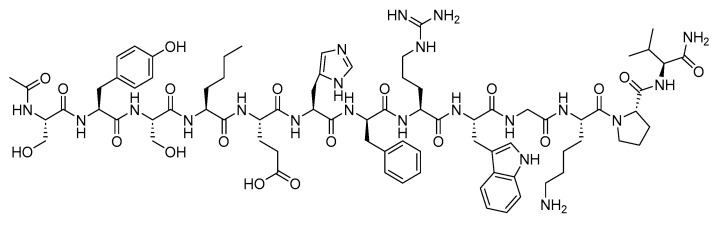
Chemical structure of afamelanotide.

**Figure 40 biomolecules-14-00264-f040:**
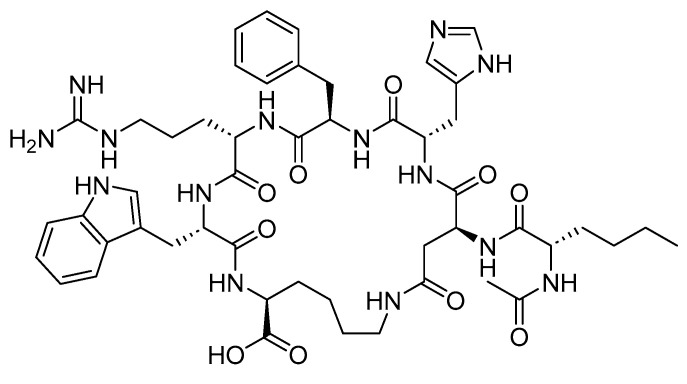
Chemical structure of bremelanotide.

**Figure 41 biomolecules-14-00264-f041:**
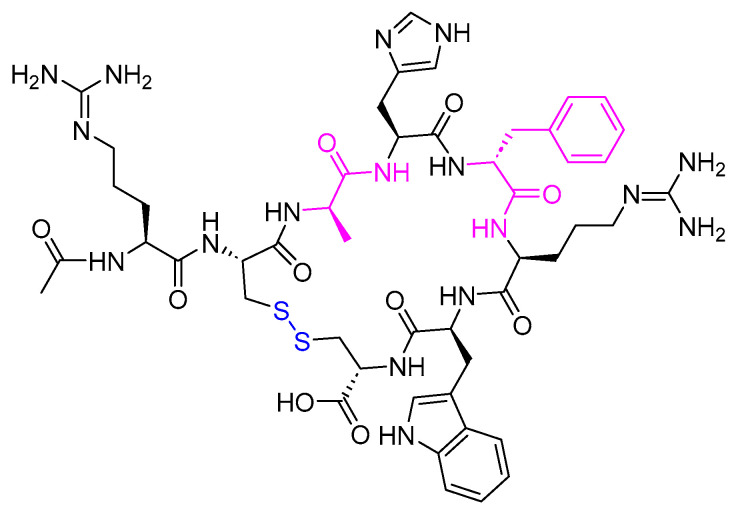
Chemical structure of setmelanotide. Blue: disulfide bridges; pink: D-amino acids.

**Table 1 biomolecules-14-00264-t001:** FDA approved natural peptides.

Peptide (Trade Name)	Indication	Therapeutic Target	Route	FDA Approval Year
Insulin (Iletin)	To treat diabetes mellitus	Insulin receptors	SC	1923
Corticotropin (H.P. Acthar)	To treat a variety of specific and inadequately defined steroid responsive disorders, multiple sclerosis, and infantile spasms in infants and children under 2 years of age	CRH-R1 and CRH-R2	SC, IM	1952
Cyclosporine (Sandimmune)	Immunosuppressant agent	Receptor cyclophilin-1	Orally	1983
Oxytocin (Syntocinon)	Uterine-contracting and milk-ejecting hormone	Protein G	IV	1996
Glucagon (Baqsimi)	To manage and treat hypoglycemia as an antidote to beta-blocker and calcium channel blocker overdose, as an anaphylaxis refractory to epinephrine, and to aid in passing food boluses	Glucagon receptor	IV, IM, or SC	1998
Secretin (ChiRhoStim)	Regulation of gastric acid, regulation of pancreatic bicarbonate, and osmoregulation	CFTR	IV	2002
Calcitonin (Miacalcin)	To control the level of calcium in the blood	CNS receptors	IV	2005
Vasopressin (Vasostrict)	To increase the blood pressure in adults with vasodilatory shock who remain hypotensive after fluids and catecholamine	Vasopressin receptor	IV	2014
Parathyroid hormone (PTH) (Natpara)	An adjunct to calcium and vitamin D to control hypocalcemia in patients with hypoparathyroidism	PTH receptor	SC	2015
Angiotensin II (Giapreza)	A vasoconstrictor to increase blood pressure in adults with septic or other distributive shock	Angiotensin II Receptor type I	IV	2017

CFTR, fibrosis transmembrane conductance regulator; CRH-R1, corticotropin releasing hormone receptor-1; CRH-R2, corticotropin releasing hormone receptor-2; CNS, central nervous system; IM, intramuscular; IV, intravenous; PTH, parathyroid hormone; SC, subcutaneous.

**Table 2 biomolecules-14-00264-t002:** FDA approved type I and II diabetes treatments.

Peptide (Trade Name)	Indication	Therapeutic Target	Route	FDA Approval Year
Insulin	To treat diabetes mellitus.	Insulin receptor		1923
Pramlintide (Symlin)	To treat diabetic patients treated with insulin (type I and II).	Calcitonin receptor, plus one of three receptor activity-modifying proteins, RAMP1, RAMP2, or RAMP3		2005
Exenatide (Byetta)	An adjunct to diet and exercise to improve glycemic control in adults with type 2 diabetes mellitus.	GLP-1 receptor	SC	2005
Liraglutide (Victoza)	1. An adjunct to diet and exercise to improve glycemic control in adults with type 2 diabetes mellitus. 2. To reduce the risk of major adverse cardiovascular events in adults with type 2 diabetes mellitus and established cardiovascular disease.			2010
Lixisenatide (Adlyxin)	An adjunct to diet and exercise to improve glycemic control in adults with type 2 diabetes mellitus.			2013
Albiglutide (Tanzeum)	To treat type 2 diabetes mellitus.			2014 *
Dulaglutide (Trulicity)	1. An adjunct to diet and exercise to improve glycemic control in adults with type 2 diabetes mellitus. 2. To reduce the risk of major adverse cardiovascular events in adults with type 2 diabetes mellitus who have established cardiovascular disease or multiple cardiovascular risk factors.			2014
Semaglutide (Ozempic)	An adjunct to diet and exercise to improve glycemic control in adults with type 2 diabetes mellitus.			2017
Tirzepatide (Mounjaro)	An adjunct to diet and exercise to improve glycemic control in adults with type 2 diabetes mellitus.	GLP-1 and GIP receptors		2022

GIP, glucose-dependent insulinotropic polypeptide; GLP-1, glucagon-like peptide-1; SC, subcutaneous. * Albiglutide was withdrawn in 2017.

**Table 3 biomolecules-14-00264-t003:** FDA approved growth-hormone-releasing hormone (GHRH) peptides.

Peptide (Trade Name)	Indication	Therapeutic Target	Route	FDA Approval Year
Sermorelin (Geref)	To reduce the excess abdominal fat in human immunodeficiency virus (HIV)-infected adult patients with lipodystrophy.	GHRH	SC	1991
Mecasermin (Increlex)	To treat growth failure in children with severe primary IGF-1 deficiency (primary IGFD) or with GH gene deletion who have developed neutralizing antibodies to GH.			2005
Tesamorelin (Egrifta)	To reduce the excess abdominal fat in HIV-infected patients with lipodystrophy.			2010
Macimorelin (Macrilen)	To diagnose adult GH deficiency.		Orally	2017

HIV, human immunodeficiency virus; SC, subcutaneous.

**Table 4 biomolecules-14-00264-t004:** FDA approved peptide-based ACTH analogues.

Peptide (Trade Name)	Indication	Therapeutic Target	Route	FDA Approval Year
Corticorelin (Acthrel)	To evaluate the status of the pituitary–adrenal axis.	Anterior pituitary	IV	1996
Corticotropin (Cosyntropin)	To diagnose patients presumed to have adrenocortical insufficiency.	Receptor in the adrenal cell plasma membrane	IV	2008

ACTH, adrenocorticotropic hormone; IV, intravenous.

**Table 5 biomolecules-14-00264-t005:** FDA approved peptide-based α-MSH analogues.

Peptide (Trade Name)	Indication	Therapeutic Target	Route	FDA Approval Year
Afamelanotide (Scenesse)	To increase pain-free light exposure in adult patients with a history of phototoxic reactions from erythropoietic protoporphyria (EPP).	MC1R	SC	2019
Bremelanotide (Vyleesi)	To treat premenopausal women with acquired, generalized hypoactive sexual desire disorder (HSDD).	MC1R and MC4R		2019
Setmelanotide (Imcivree)	To treat chronic weight management in adult and pediatric patients.	MC4R		2020

α-MSH, α-Melanocyte stimulating hormone; SC, subcutaneous; MC1R, melanocortin 1 receptor; MC4R, melanocortin 4 receptor.

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
