# Peer review of "Exploring FDA-Approved Frontiers: Insights into Natural and Engineered Peptide Analogues in the GLP-1, GIP, GHRH, CCK, ACTH, and α-MSH Realms"

_biomolecules, 2024, doi:10.3390/biom14030264_

Round 1

Reviewer 1 Report

Comments and Suggestions for Authors

This review examines FDA-approved peptide drugs, detailing their structural diversity, therapeutic targets, and mechanisms of action. The insights provided can significantly inform future advancements in peptide drug synthesis. I recommend publication in Biomolecules without modification, except for a minor correction at line 46: "...market [1, 9]. Cyclisation is a prevalent chemical modification employed to restrict the structure of peptides...", where the redundant "is a" should be removed.

Author Response

I attached my response

Reviewer 2 Report

Comments and Suggestions for Authors

The author attempts to summarize FDA-approved natural and engineered peptide analogues in GLP-1, GIP, GHRH, CCK, ACTH, and α-MSH. However, simply listing these approved drugs feels inadequate. The specific pharmacological characteristics, metabolism within the body, and mechanisms of action of these drugs are not thoroughly explained, nor is there adequate cross-comparison of these approved peptides. I believe a comprehensive review should not only list but also compare these peptides in terms of pharmacology, structure, and mechanism of action, providing new insights and guidance for future developments. In this regard, the article falls short and needs improvement.

I have the following recommendations:

1.      The pharmacology and mechanisms of action for each peptide analogue should be described in detail.

2.      ACTH was first approved in 1952.

3.      Pasireotide (Signifor) may not be an analogue of ACTH.

4.      Some descriptions of Bremelanotide might be inaccurate. Pharmacology and mechanisms should be described. LIN3-668, an MC1R agonist, should be corrected to MCR agonist.

5.      Setmelanotide, an FDA-approved MC4R agonist for treating obesity since 2020, should be included in this paper.

Author Response

I attached my response

Reviewer 3 Report

Comments and Suggestions for Authors

It would be beneficial to include a figure that elucidates the mechanisms of action of the different peptides discussed, providing readers with more comprehensive information.

Regarding the engineered peptide analogues, the authors could delve into specific modifications made to peptides aimed at enhancing their stability and efficacy, thereby enriching the discussion.

Expanding the conclusion to include more insights on the future direction of peptide drug development would provide a more comprehensive overview of the topic.

Comments on the Quality of English Language

In the first paragraph of the Introduction, it would be beneficial to include the full names of CSPS, LPPS, and SPPS for clarity.

Please correct the typo identified in line 46.

Author Response

I attached my response

Round 2

Reviewer 2 Report

Comments and Suggestions for Authors

The authors have included the corrections, so it can be accepted for publication